# Age-related changes in 'cortical' 1/f dynamics are linked to cardiac activity

Fabian Schmidt[1]*, Sarah K Danböck[1], Eugen Trinka[1,2,3], Dominic P Klein[4], Gianpaolo Demarchi[1], Nathan Weisz[1,2]

[1]Paris-Lodron-University of Salzburg, Department of Psychology, Centre for Cognitive Neuroscience, Salzburg, Austria; [2]Neuroscience Institute, Christian Doppler University Hospital, Paracelsus Medical University, Salzburg, Austria; [3]Department of Neurology, Christian Doppler University Hospital, Paracelsus Medical University, Salzburg, Austria; [4]Division of Cardiology and Emergency Medicine, Department of Medicine V, Clinic Favoriten, Vienna, Austria

*For correspondence:
Fabian.Schmidt@plus.ac.at

Competing interest: The authors declare that no competing interests exist.

## eLife Assessment

Examination of (a)periodic brain activity has gained particular interest in the last few years in the neuroscience fields relating to cognition, disorders, and brain states. Using large EEG/MEG datasets from younger and older adults, the current study provides **compelling** evidence that age-related differences in aperiodic EEG/MEG signals can be driven by cardiac rather than brain activity. Their findings have **important** implications for all future research that aims to assess aperiodic neural activity, suggesting control for the influence of cardiac signals is essential.

**Abstract** The power of electrophysiologically measured cortical activity decays with an approximately $1/f^X$ function. The slope of this decay (i.e. the spectral exponent, $X$) is modulated by various factors such as age, cognitive states or psychiatric/neurological disorders. Interestingly, a mostly parallel line of research has also uncovered similar effects for the spectral slope in the electrocardiogram (ECG). This raises the question of whether these bodywide changes in spectral slopes are (in-)dependent. Focusing on well-established age-related changes in spectral slopes, we analyzed a total of 1282 recordings of magnetoencephalography (MEG) resting state measurements with concurrent ECG in an age-diverse sample (18–88 years). Using a diverse array of analytical approaches, we demonstrate that the aperiodic signal recorded via surface electrodes/sensors originates from multiple physiological sources. Furthermore, our results suggest that common 'artifact' rejection approaches (i.e. ICA) may not be sufficient to separate cardiac from neural activity. In particular, significant parts of age-related changes in aperiodic activity normally interpreted to be of neural origin can be explained by cardiac activity. Moreover, our results suggest that changes (flattening/steepening) of the spectral slope with age are dependent on the recording site and investigated frequency range. Our results highlight the complexity of aperiodic activity while raising concerns when interpreting aperiodic activity as 'cortical' without considering physiological influences.

## Introduction

Aperiodic neural activity is omnipresent both in invasive (e.g. ECoG *Miller et al., 2009*) and non-invasive (e.g. MEG/EEG *Demanuele et al., 2007*; *Pritchard, 1992*) recordings of electrophysiological brain activity and even in hemodynamic responses (e.g. fMRI *He et al., 2010*). In the frequency domain, when visualized in log-log coordinates (log-frequency/log-power), aperiodic activity manifests as a linear decay in power with an increase in frequency (*Miller et al., 2009*) (i.e. the spectral

slope). This part of the signal - following a so-called 'power-law' distribution - is often referred to as 'scale-free', '1 /f noise' or more recently 'aperiodic activity' (*Donoghue et al., 2020*; see *Figure 1A* for an illustration of aperiodic activity with different spectral slopes in the time and frequency domain). In the past, aperiodic neural activity was often treated as noise and simply removed from the signal, for example via pre-whitening (*Buzsáki, 2006*; *Mitra and Pesaran, 1999*), so that analyses could focus on periodic neural activity (local peaks that rise above the 'power-law' distribution, which are commonly thought to reflect neural oscillations). However, in recent years, the analysis of aperiodic neural activity has gained interest (see *Figure 1D*). Several studies have shown that aperiodic neural activity is meaningfully modulated by various factors, such as age (*Voytek et al., 2015*), cognitive state (e.g. awake vs. sleep *He et al., 2010*) and disorders like Parkinson's disease and epilepsy (*Ghinda et al., 2020*; *Mostile et al., 2019*). However, aperiodic activity is not only present in recordings of neural activity, but also part of other physiological signals such as cardiac and muscle activity, commonly measured using electrocardiography (ECG *Saul et al., 1988*) and electromyography (*Kozhemiako et al., 2022*). Interestingly, and mostly overlooked by the neuroscience community (see *Figure 1C*), aperiodic activity measured using ECG (often referred to as power law or 1/f activity) is modulated by similar factors as neural aperiodic activity, including aging (*Beckers et al., 2006*), different cognitive states (e.g. awake vs. sleep *Kozhemiako et al., 2022*; *Penzel et al., 2003*) and disorders such as Parkinson's disease and epilepsy (*Ansakorpi et al., 2002*; *Haapaniemi et al., 2001*; see also *Figure 1B*). Furthermore, it is well-known that, via volume conduction, cardiac activity can also be captured in both invasive and non-invasive recordings of neural activity (*Dirlich et al., 1997*; *Jousmäki and Hari, 1996*; *Kern et al., 2013*). Hence, it is also considered best practice to measure and remove cardiac activity from M/EEG recordings (*Gross et al., 2013*). However, an analysis of openly accessible M/EEG articles that investigate aperiodic activity ($N_{Articles}$ = 279; see Methods - *Literature Analysis* for further details) revealed that only 17.1% of EEG studies explicitly mention that cardiac activity was removed and only 16.5% measure ECG (45.9% of MEG studies removed cardiac activity and 31.1% of MEG studies mention that ECG was measured; see *Figure 1E,F*). Additionally, investigations of aperiodic activity vary strongly by both the upper and lower bounds and the general width of the analyzed frequency ranges (*Figure 1G-I*). This further complicates the comparison of results across studies as physiological signals (e.g. cardiac activity) may have varying impacts across frequency ranges—for instance, exerting a stronger influence on lower frequencies than on higher ones.

Considering that (A) aperiodic neural and cardiac activity are modulated by similar traits, states, and disorders, and (B) cardiac activity is often present but rarely removed from neural recordings, we ask: Are changes in aperiodic neural activity (in-)dependent from changes in aperiodic cardiac activity? To address this question, we turn our attention to the recently reported (*Donoghue et al., 2020*; *Voytek et al., 2015*) and replicated (*Merkin et al., 2023*) association between aperiodic activity and chronological age. Using the publicly available CamCAN dataset (*Shafto et al., 2014*; *Taylor et al., 2017*), we find that the aperiodic signal measured using M/EEG originates from multiple physiological sources. In particular, significant portions of age-related changes in aperiodic activity – normally attributed to neural processes – can be better explained by cardiac activity. This observation holds across a wide range of processing options and control analyses (see *Control Analyses: Age-related steepening of the spectral slope in the MEG*), and was replicable on a separate MEG dataset. However, the extent to which cardiac activity accounts for age-related changes in aperiodic activity varies with the investigated frequency range and recording site. Importantly, in some frequency ranges and sensor locations, age-related changes in neural aperiodic activity still prevail. But does the influence of cardiac activity on the aperiodic spectrum extend beyond age? In a preliminary analysis, we demonstrate that working memory load modulates the aperiodic spectrum of 'pure' ECG recordings. The direction of this working memory effect mirrors previous findings on EEG data (*Donoghue et al., 2020*) suggesting that the impact of cardiac activity goes well beyond aging. In sum, our results highlight the complexity of aperiodic activity while cautioning against interpreting it as solely 'neural' without considering physiological influences.

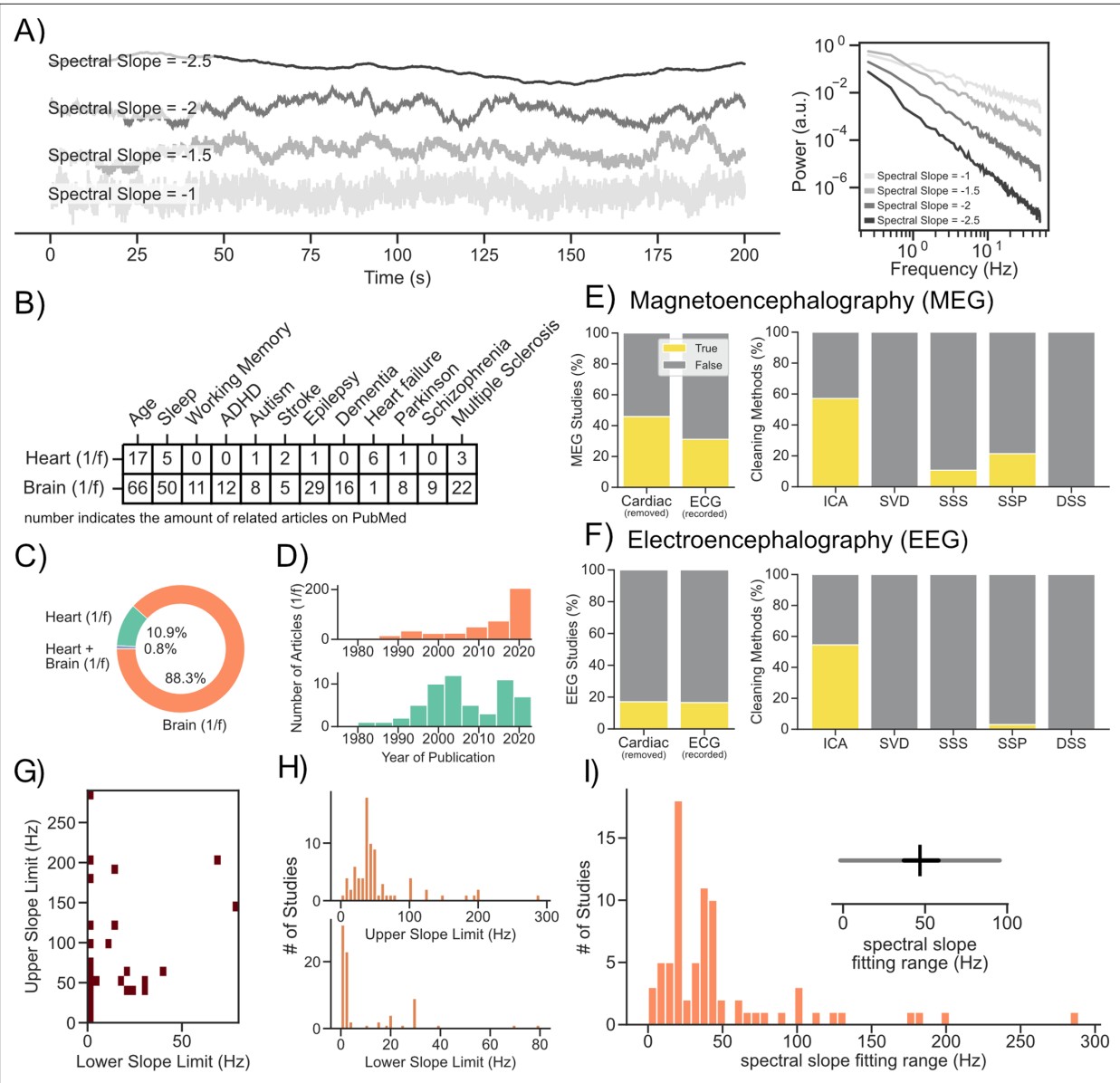

**Figure 1.** Literature analysis of aperiodic activity investigated using M/EEG and ECG. (**A**) Illustration of different types of aperiodic activity in the time and frequency domain. (**B, C**) We analyzed 489 abstracts indexed on PubMed using LISC (**Donoghue, 2019**), a package for collecting and analyzing scientific literature. (**B**) This analysis revealed that changes in aperiodic activity are related to similar traits, states, and disorders in measures of both neural and cardiac activity. (**C**) We further noted a tiny overlap of studies (N=4) that refer to both cardiac and cortical aperiodic activity in their abstracts. Yet, none of these studies considers confounding influences of cardiac aperiodic activity on the measurement of cortical aperiodic activity. (**D**) We additionally found a steep increase related to the investigation of neural aperiodic activity in the 2020s highlighting the current interest of the topic in the neuroscience community. (**E, F**) We further downloaded and analyzed freely available full texts of M/EEG studies investigating aperiodic activity to see to which extent and how cardiac activity was handled. This analysis revealed that only 17.1% of EEG studies remove cardiac activity and only 16.5% measure ECG (for MEG 45.9% removed cardiac; 31.1% mention ECG was measured). We were further interested in determining which artifact rejection approaches were most commonly used to remove cardiac activity, such as independent component analysis (ICA; **Hyvärinen, 1999**), singular value decomposition (SVD; **Lagerlund et al., 1997**), signal space separation (SSS; **Taulu and Simola, 2006**), signal space projections (SSP **Uusitalo and Ilmoniemi, 1997**), and denoising source separation (DSS; **de Cheveigné, 2010**). We found that the most commonly applied method both in EEG and MEG recordings was independent component analysis (ICA). (**G, H**) An arbitrary selection of previous studies (N=60) shows a vast amount of different frequency ranges used to investigate aperiodic activity. While a significant amount of studies looked into a range between ~0.1 and 50 Hz (~30%), most studies used a unique frequency range. (**I**) Not only do the upper and lower bounds vary between studies, but also the general width of the fitting range can vary from 0.9 to 290 Hz.

# Results

## Aperiodic signals recorded using ECG are associated with aging and heart rate variability

Changes of aperiodic activity in recordings of neural and cardiac activity are associated with aging (*Voytek et al., 2015*; *Beckers et al., 2006*). However, analyses of ECG signals - in the frequency domain - typically focus on (a-)periodic signals <0.4 Hz (*Shaffer and Ginsberg, 2017*). These (compared to neural time series) slowly fluctuating signals are related to heart rate variability (*Task Force of the European Society of Cardiology the North American Society of Pacing Electrophysiology, 1996*). Changes in (a-)periodic activity at these low frequencies are well-established physiological measures (*Azuaje et al., 2006*). However, substantially less is known about aperiodic activity above 0.4 Hz in the ECG. Yet, common ECG setups for adults capture activity at a broad bandwidth of 0.05–150Hz (*Tereshchenko and Josephson, 2015*; *Kusayama et al., 2020*). Importantly, a lot of the physiological meaningful spectral information rests between 1 and 50 Hz (*Azuaje et al., 2006*), similarly to M/EEG recordings. Furthermore, meaningful information can be extracted at much higher frequencies. For instance, ventricular late potentials have a broader frequency band (~40–250 Hz *Azuaje et al., 2006*). However, that's not all, as further meaningful information can be extracted at even higher frequencies (>100 Hz). For instance, the so-called high-frequency QRS seems to be highly informative for the early detection of myocardial ischemia and other cardiac abnormalities that may not yet be evident in the standard frequency range (*Schlegel et al., 2004*; *Qiu et al., 2024*). Yet, the exact physiological mechanisms underlying the high-frequency QRS remain unclear (see *Qiu et al., 2024* for a review discussing possible mechanisms).

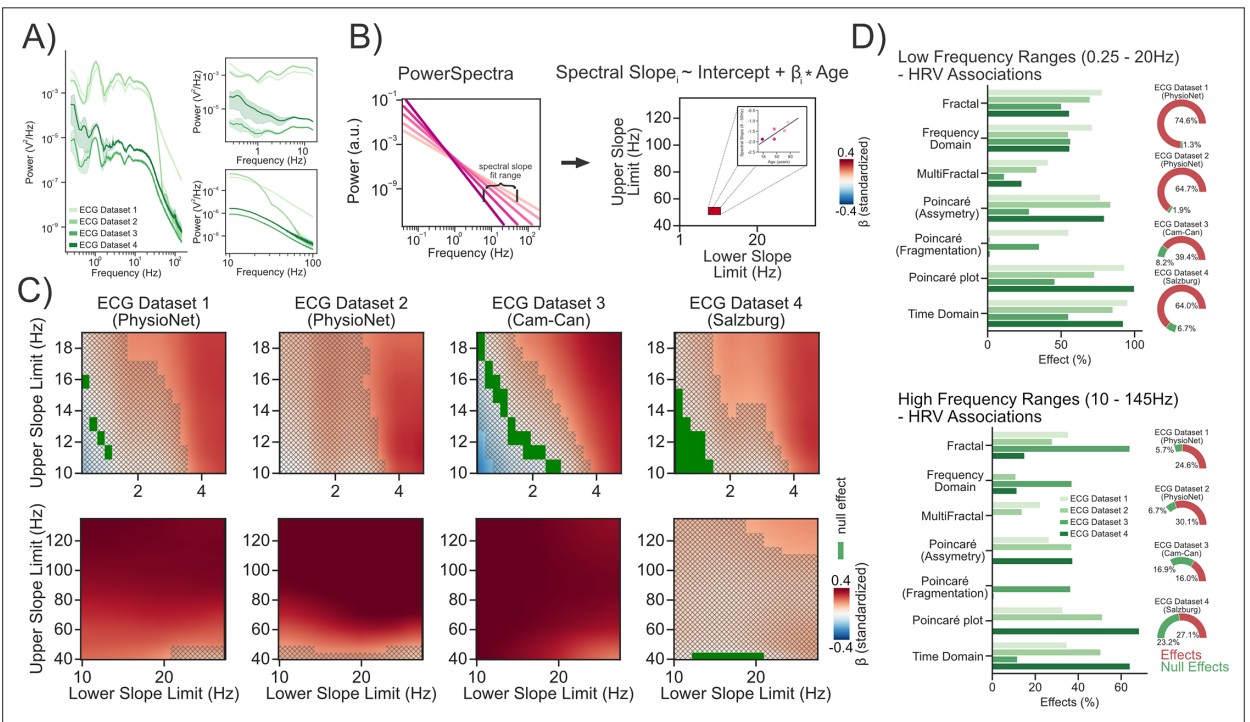

**Figure 2.** Aperiodic signals recorded using ECG are related to aging and heart rate variability. (**A**) Grand average power spectra plotted separately per Dataset and the associated aperiodic power spectra for the lower (0.25–20 Hz) and higher (10–145 Hz) frequency ranges. (**B, C**) Age was used to predict the spectral slope using different upper and lower slope limits for higher (10–145 Hz) and lower (0.25–20 Hz) frequency ranges. Significant effects, that is effects with credible intervals not overlapping with a region of practical equivalence (ROPE; see Methods - Statistical Inference), are highlighted in red or blue (see colorbar). Null effects, which were defined as effects with credible intervals completely within a ROPE, are highlighted in green. Results where no decision to accept or reject (see *Kruschke, 2018*) an effect could be made are masked using hatches. (**D**) To understand whether aperiodic cardiac activity also relates to common measures of heart rate variability, we predicted the spectral slope using 90 different measures of heart rate variability. We find consistent (yet different) associations with mostly fractal and time domain measures in both lower and higher frequency ranges.

The online version of this article includes the following figure supplement(s) for figure 2:

**Figure supplement 1.** ECG spectra and knee frequency: aperiodic signals recorded using ECG can be associated with aging.

To understand whether aperiodic activity recorded using ECG carries meaningful information about aging – at frequency ranges common in M/EEG recordings– the ECG data of 4 age-diverse populations with a total of 2286 subjects were analyzed. After pre-processing (see Methods), age was used to predict the spectral slope of the ECG over various different frequency ranges (see *Figure 2C*). Due to the presence of a 'knee' in the ECG data (for details regarding 'knees' in power spectra see *Miller et al., 2009*; *Donoghue et al., 2020*; *Gao et al., 2020*), slopes were fitted individually to each subject's power spectrum in several lower (0.25–20 Hz) and a higher (10–145 Hz) frequency ranges. The split in lower and higher frequency ranges was performed to avoid spectral knees at ~15 Hz in the center of the slope fitting range (see *Figure 2—figure supplement 1B* for the distribution of knee frequencies across datasets). Our results show that the spectral slope flattened with age over a vast amount of different frequency ranges (see *Figure 2C*). These results are similar to what was reported in previous studies measuring 'cortical' aperiodic activity. However, we also noted an age-related steepening of the spectral slope in one dataset (ECG Dataset 3 - CamCAN) in the low-frequency range (0.25–12 Hz, see Discussion).

But do these aperiodic changes at the ECG also correspond to established indices of cardiac health and function? To better understand this, we conducted an exploratory analysis, where we related the spectral slope of the aperiodic ECG signal in various frequency ranges to 90 different indices of heart rate variability (implemented in NeuroKit2 *Makowski et al., 2021*) across all four datasets. The results show that spectral slopes both in lower (0.25–20 Hz) and higher (10–145 Hz) frequency ranges relate to several indices of heart rate variability (see *Figure 2D*). Overall, spectral slopes in lower frequency ranges were more consistently related to heart rate variability indices (39.4–74.6% percent of all investigated indices) than spectral slopes in higher frequency ranges (16–30.01% percent of all investigated indices; see *Figure 2D*). In the lower frequency ranges (0.25–20 Hz), spectral slopes were consistently related to most measures of heart rate variability; that is significant effects were detected in all four datasets (see *Figure 2D*). This includes fractal, multifractal, time and frequency domain analyses as well as indices extracted from the Poincaré plot. In the higher frequency ranges (10–145 Hz), spectral slopes were most consistently related to fractal and time domain indices of heart rate variability, but not so much to frequency-domain indices assessing spectral power in frequency ranges <0.4 Hz. This suggests that spectral slopes >10 Hz carry meaningful information about cardiac activity that is largely distinct from the frequency-domain information that is commonly investigated using ECG. In sum, these findings show that aperiodic activity, in frequency ranges that vastly exceeds those commonly explored in ECG analyses, may carry meaningful information about cardiac activity.

With regard to aging, the conducted analyses show that aperiodic activity measured via ECG is associated with aging at frequency ranges vastly exceeding those typically investigated via ECG, but overlapping with frequency ranges commonly measured in recordings of neural activity (see *Figure 1G*). Importantly, the direction of the association between age and aperiodic ECG activity is largely identical to that reported for age and aperiodic EEG activity (*Donoghue et al., 2020*; *Voytek et al., 2015*), motivating the investigation of these relationships in combined neural and cardiac measurements.

## Cardiac activity is directly captured in EEG and MEG recordings

Aperiodic activity recorded using ECG (see *Figure 2C*) and EEG/ECoG (*Voytek et al., 2015*) is similarly modulated by age. In MEG and some EEG recordings, cardiac activity is measured via ECG (*Gross et al., 2013*). Components of the signal related to cardiac activity are then commonly removed via independent component analysis (ICA *Hyvärinen, 1999*; see *Figure 1E,F*). In recordings of EEG, the influence of cardiac activity is often deemed less problematic *Dirlich et al., 1997*; as a result, ECG is rarely recorded (see *Figure 1E*). We utilized concurrent ECG, EEG, and MEG resting state recordings to examine to what extent ECG signals are present in the signals measured using MEG and EEG. We calculated so-called temporal response functions (see Methods), to detect whether the signals recorded at different locations (M/EEG vs. ECG) are instantaneously related (zero-time-lags; and therefore likely correspond to the same source) or if one lags behind the other (non zero-time-lags; likely different sources influencing each other, for example via interoception see *Schulz, 2016*). After pre-processing (see Methods), the data was split into three conditions using an ICA *Hyvärinen, 1999*. Independent components that were correlated (at *r*>0.4; see Methods: *MEG/EEG Processing - pre-processing*) with the ECG electrode were either not removed from the data (*Figure 3A-D* - blue),

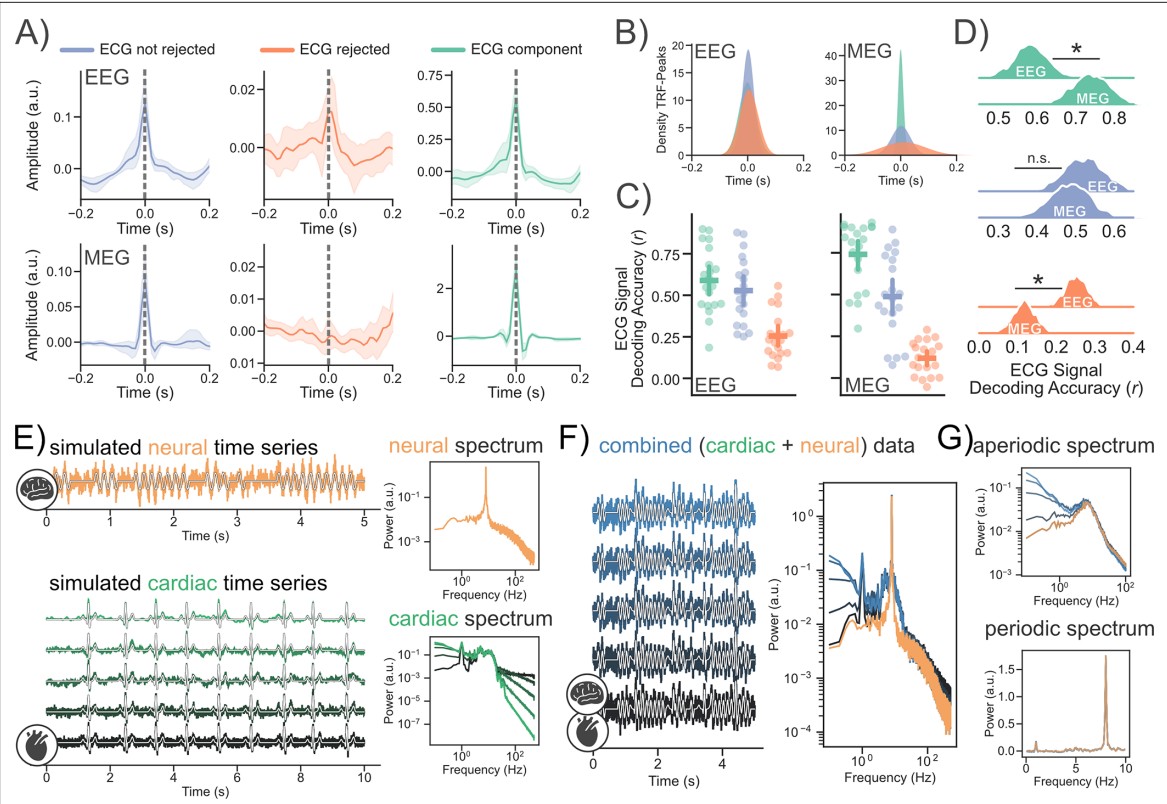

**Figure 3.** Cardiac activity is captured in EEG and MEG recordings. (**A, B**) Cardiac activity is captured at zero time lags in concurrent MEG and EEG recordings. If the ECG signal is rejected via ICA, this effect disappears in MEG, but not completely in EEG data. (**C, D**) Reconstruction of the ECG signal was impaired, but remained possible even after rejecting the ICA signal using ECG (both in MEG and EEG data). Notably, reconstruction of the ECG signal (after ICA) worked better in EEG than MEG data. A * indicates a 'significant' effect (see Methods - Statistical Inference). Error bars indicate the 95% confidence interval of the mean. (**E**) To illustrate how aperiodic activity recorded using ECG might impact neural aperiodic activity, we simulated cardiac and neural time series data. The neural time series data was simulated as in *Gao et al., 2017* with an EI ratio of 1:2. The cardiac time series consists of a PQRST-Complex and different types of 1 /f noise. (**F**) Combining both cardiac and neural time series data shows that even if the PQRST-Complex is barely visible in the combined time domain signal, the resulting power spectrum can be heavily affected by simulated changes in aperiodic cardiac activity (**F, G**).

removed from the data (*Figure 3A-D* - orange) or projected back into the sensor space (*Figure 3A-D* - green). Afterward, temporal response functions (encoding models; see Methods) between the signal recorded at the ECG electrode and the MEG/EEG sensors and feature reconstruction models (decoding models) were computed (for each condition, respectively). Our results show that if ECG components are not removed via ICA, the ECG signal is captured equally strong at zero-time lags both in EEG and MEG recordings (see *Figure 3A,C,D*). Even after removing ECG-related components from the data, TRF peaks emerged (although reduced) at zero-time lags in EEG, but not in MEG recordings (see *Figure 3A,B*). Furthermore, reconstruction (decoding) of the ECG signal was reduced, but remained above chance even after rejecting the ICA signal using ECG both in MEG and EEG recordings ($r>0$). Interestingly, the presence of the ECG signal was more pronounced in EEG compared to MEG recordings, after removing ECG-related components ($\beta_{standardized\ (EEG > MEG)}$=0.97, HDI = [0.42, 1.52]; *Figure 3D*). Additionally, ECG-related components extracted from MEG recordings were more related to the ECG signal than the components extracted from the EEG ($\beta_{standardized\ (EEG > MEG)}$=−0.76, HDI = [-1.35,−0.18]; *Figure 3D*). These results show that (A) residual ECG activity remains in surface neural recordings, even after applying a very sensitive threshold to detect and remove ECG components via ICA and (B) neural and cardiac activity are more difficult to separate in EEG as opposed to MEG recordings (see *Figure 3A,C,D*), resulting in more residual (after ICA) ECG-related activity in EEG recordings. To further illustrate how changes in aperiodic cardiac activity might impact 'cortical' aperiodic activity recorded via M/EEG, we simulated cardiac and neural time series data (see *Figure 3E*). The neural time series data was simulated as in *Gao et al., 2017* with an EI ratio of 1:2. The cardiac

time series consists of a simulated template PQRST-Complex at a rate of ~1 Hz (with jittered onsets) and different types of additional aperiodic activity. Combining both cardiac and neural time series data shows that even if the PQRST-Complex is barely visible in the combined time domain signal, the resulting power spectrum can be heavily affected by simulated changes in aperiodic cardiac activity (see *Figure 3F,G*).

## Age-related changes in aperiodic brain activity are most pronounced in cardiac components

ECG signals are captured in brain activity recorded using M/EEG (see *Figure 3A-D*). Furthermore, aperiodic activity recorded using ECG is –just like aperiodic activity recorded using EEG/ECoG– modulated by age (see *Figure 2C*). However, it is unclear whether these bodywide changes in aperiodic activity are (in-)dependent. To answer this question, we are leveraging resting state MEG recordings of an age-diverse population obtained from the CamCAN inventory (N=627 *Shafto et al., 2014*; *Taylor et al., 2017*). After pre-processing (see Methods), an ICA was applied to separate MEG activity from activity related to the ECG. ICA components that were related to the ECG signal were identified using a correlation threshold ($r$>0.4; same threshold as in 3ABCD). The data was split into three conditions ($MEG_{ECG\ not\ rejected}$, $MEG_{ECG\ rejected}$, and $MEG_{ECG\ component}$; see *Figure 3A*) and projected back to the sensor space, respectively. Age was then used to predict the spectral slope across 102 magnetometers and over a wide variety of frequency ranges with lower limits starting at 0.5 Hz in 1 Hz steps ranging until 10 Hz and upper limits starting at 45 Hz in 5 Hz steps ranging until 145 Hz (see *Figure 4B*) per subject (split by condition). This analysis, which is depicted in *Figure 4*, shows that over a broad amount of individual fitting ranges and sensors, aging resulted in a steepening of spectral slopes across conditions (see *Figure 4B*) with 'steepening effects' observed in 25% of the slope fits in $MEG_{ECG\ not\ rejected}$, 0.5% in $MEG_{ECG\ rejected}$, and 60% for $MEG_{ECG\ components}$. The second largest category of effects was 'null effects' in 13% of the options for $MEG_{ECG\ not\ rejected}$, 30% in $MEG_{ECG\ rejected}$, and 7% for $MEG_{ECG\ components}$. However, we also found 'flattening effects' in the spectral slope for 0.16% of the processing options in $MEG_{ECG\ not\ rejected}$, 0.3% in $MEG_{ECG\ rejected}$, and 0.46% in $MEG_{ECG\ components}$.

Interestingly, this analysis shows that overall options, both flattening and steepening effects were most frequently observed on the $MEG_{ECG\ components}$. This analysis also indicates that a vast majority of observed effects, irrespective of condition (ECG components, ECG not rejected, ECG rejected), show a steepening of the spectral slope with age across sensors and frequency ranges. This finding is contrary to previous findings showing a flattening of spectral slopes with age in recordings of both brain (*Voytek et al., 2015*) and cardiac activity (*Beckers et al., 2006*) (see also *Figure 2C*). We therefore conducted several control analyses both on data averaged across sensors (see *Figure 4—figure supplements 1–6*) and on the level of single sensors (see *Figure 4—figure supplements 7 and 8*) to investigate to what degree this observation is based on decisions made during preprocessing (see *Control Analyses: Age-related steepening of the spectral slope in the MEG*). In sum, all performed control analyses indicate that aging can robustly cause a steepening of the spectral slope in "cortical" activity recorded using MEG that is not explainable by age-related changes in head movements or EOG activity, the application of different blind source separation algorithms (e.g. ICA and SSS), the algorithm used to extract the spectral slope (IRASA vs. FOOOF), and replicable across two large MEG datasets. These steepening effects have previously not been reported in EEG recordings, which suggests that they may be in part linked to physiologically measured 1 /f noise differently affecting magnetic and electric recording devices (*Dehghani et al., 2010*; see Discussion). However, despite the large amount of age-related steepening effects, we also noted age-related flattening in spectral slopes that occurred mainly at centrally and parietally located electrodes in lower frequency ranges between 0.5 and 45 Hz (see *Figure 4C*). Importantly, these results overlap both in frequency range and recording site with some of the results previously reported in the literature (*Voytek et al., 2015*; *Tröndle et al., 2022*). A majority of results fall in the category 'undecided' (see *Figure 4E-G*) as there was not enough evidence to either support a steepening, flattening, or null effect (*Kruschke, 2018*). Albeit undecided, we still visualized the respective direction of these results labeling them either as 'undecided/steepening' or 'undecided/flattening' to give a descriptive overview of the associated spatial locations and frequency ranges in which these results were observed (see *Figure 4E,F*). In sum, this analysis suggests that a flattening of spectral slopes with age can be observed at some of the previously reported

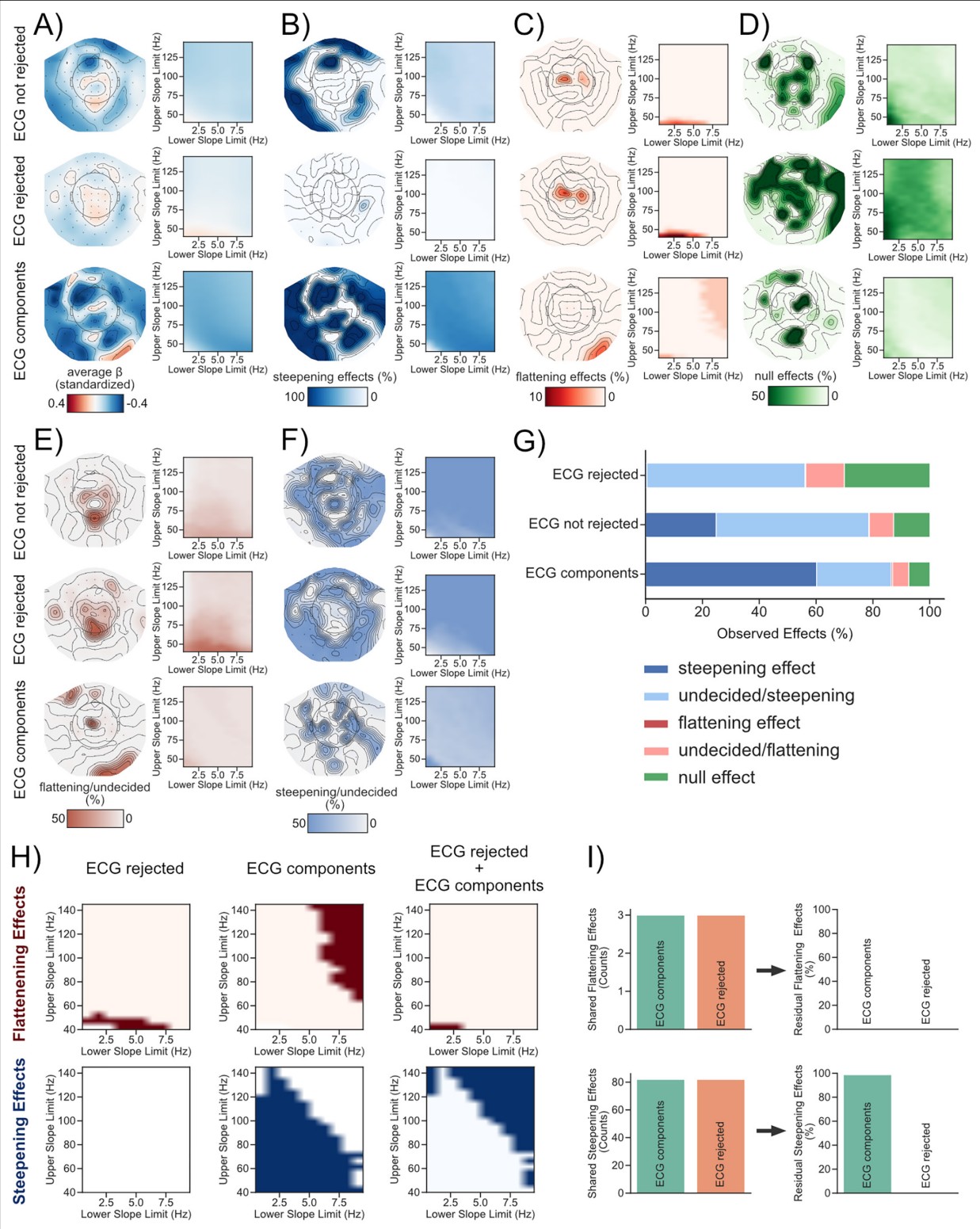

**Figure 4.** Age-related changes in aperiodic brain activity are most pronounced in cardiac components. Age was used to predict the spectral slope at rest in three different conditions (ECG components not rejected, ECG components rejected, and ECG components only) per channel across a variety of frequency ranges. (**A**) Standardized beta coefficients either per channel averaged across all frequency ranges (left) or per frequency range (right) averaged across all channels. Age-related (**B**) steepening, (**C**) flattening, and (**D**) null effects in the spectral slope were observed and visualized in a similar manner as in (**A**). (**E, F**) We further show the direction of results where we did not find enough evidence to support either a steepening, flattening, or null effect. (**G**) Summary of all observed findings in %. (**H**) At some frequency ranges, neural and cardiac aperiodic activity change independently with

*Figure 4 continued on next page*

*Figure 4 continued*

age (see also **B, C**). (**H, I**) At other frequency ranges, cardiac and neural aperiodic activity are similarly modulated by age. We used cardiac and neural aperiodic activity as predictors for age in a multiple regression model to test whether both explain unique variance in aging. This analysis reveals that when adding both MEG$_{ECG\ component}$ and MEG$_{ECG\ rejected}$ as predictors, age-related flattening effects were reduced, yielding no longer significant flattening results (**I**; upper panel). However, in the case of the observed steepening effects, significant effects for MEG$_{ECG\ components}$ remained in 98.75% of the tested frequency ranges (**I**; lower panel).

The online version of this article includes the following figure supplement(s) for figure 4:

**Figure supplement 1.** Power spectra comparison and goodness of fit metrics: Grand average power spectra for the MEG data recorded at Cambridge and split in the three conditions MEG$_{ECG\ not\ rejected}$, MEG$_{ECG\ rejected}$, MEG$_{ECG\ components}$.

**Figure supplement 2.** Replication FOOOF: age-related changes in aperiodic brain activity can be explained by cardiac components.

**Figure supplement 3.** Comparison between Fixed and Knee model fits using FOOOF.

**Figure supplement 4.** Replication Salzburg sample: age-related changes in aperiodic brain activity can be explained by cardiac components.

**Figure supplement 5.** SSS Maxfilter analysis 1: age-related changes in aperiodic brain activity are most prominent on cardiac components irrespective of maxfiltering the data using signal space separation (SSS) or not.

**Figure supplement 6.** Effects of varying ICA thresholds.

**Figure supplement 7.** SSS Maxfilter analysis 2: steepening and flattening of the spectral slope with age is dependent on the sensor location and the investigated frequency range.

**Figure supplement 8.** Head movement and EOG control analysis: steepening and flattening of the spectral slope with age is dependent on the recording site and the investigated frequency range, when controlling for head movements.

frequency ranges (~0.5–45 Hz) and spatial locations (on central, parietal, and occipital sensors). However, these results represented only 0.3% of effects across all processing settings (conditions, sensors, and frequency ranges). Even when restrictively looking only at the investigated frequency ranges between 0.5 and 50 Hz, only 1.2% (0.4% after maxfilter see *Figure 4—figure supplement 7*) of effects across these residual settings were showing an age-related flattening of the spectral slope. This suggests that age-related flattening of the spectral slope is tied to specific recording sites and frequency ranges.

## Age-related changes in aperiodic brain activity are linked to cardiac activity in a frequency-dependent manner

So far, we have shown that age-related steepening/flattening of the spectral slope in the MEG is both dependent on the investigated frequency range and the sensor selection. While a vast majority of our results indicate an age-related steepening of the spectral slope (in contrast to previous findings), we also noted a flattening of the spectral slope at a subset of central sensors in the lower frequency range (~0.5–45 Hz; in line with previous findings; *Voytek et al., 2015*; *Tröndle et al., 2022*). Some of the observed age-related flattening and steepening effects were solely present in one of the tested conditions (see *Figure 4H*). This suggests that aperiodic brain activity (MEG$_{ECG\ rejected}$), at some frequency ranges, changes with age independently of cardiac activity (MEG$_{ECG\ component}$) and vice versa. However, we also noted shared effects at other frequency ranges (i.e. effects present both in the MEG$_{ECG\ rejected}$ and MEG$_{ECG\ component}$ condition; see *Figure 4I*).

To see if MEG$_{ECG\ rejected}$ and MEG$_{ECG\ component}$ explain unique variance in aging at frequency ranges where we noticed shared effects, we averaged the spectral slope across significant channels and calculated a multiple regression model with MEG$_{ECG\ component}$ and MEG$_{ECG\ rejected}$ as predictors for age (to statistically control for the effect of MEG$_{ECG\ components}$ and MEG$_{ECG\ rejected}$ on age). This analysis was performed to understand whether the observed shared age-related effects (MEG$_{ECG\ rejected}$ and MEG$_{ECG\ component}$) are (in-)dependent. The analysis revealed that when adding both MEG$_{ECG\ component}$ and MEG$_{ECG\ rejected}$ as predictors, age-related flattening effects were reduced, yielding no longer significant flattening results (see *Figure 4B*; upper panel). However, in case of the observed steepening effects, significant effects for MEG$_{ECG\ components}$ remained in 98.75% of the tested frequency ranges (see *Figure 4B*; lower panel). In sum, these results suggest that whether or not aperiodic brain activity changes independently from cardiac activity with age depends on the recording site and the selected frequency range.

# Control analyses: age-related steepening of the spectral slope in the MEG

First, we conducted a median split of age to compare the raw power spectra averaged across channels (see *Figure 4—figure supplement 1*). This shows that on the grand average across channels, the spectral slope was slightly steeper in older subjects even before spectral parametrization. Furthermore, the use of blind source separation artifact rejection approaches may influence power spectral densities by reducing external noise in the signal. We therefore investigated whether and how the use of a Signal-Space-Separation algorithm (SSS *Taulu and Simola, 2006*; *Taulu and Kajola, 2005*) and different ICA thresholds influence the reported results. On the grand average across sensors, slightly stronger steepening effects were observed for the MEG$_{ECG not rejected}$ data compared to MEG$_{ECG rejected}$ when not cleaning the data using SSS and vice versa (see *Figure 4—figure supplement 5* for a comparison). On the level of single sensors, the application of SSS resulted in a further reduction of both flattening and steepening effects in all conditions except for MEG$_{ECG components}$, where we noted a 20% increase in steepening effects (see *Figure 4—figure supplement 7*). Considering that we detected less and weaker aperiodic effects when using SSS maxfilter, is it now advisable to omit maxfilter when analyzing aperiodic signals? We do not think that we can make such a judgment based on our current results. This is because it is unclear whether or not the reduction of effects stems from an additional removal of peripheral information (e.g. muscle activity; that may be correlated with aging) or is induced by the SSS maxfiltering procedure itself. As the use of maxfilter in detecting changes of aperiodic activity was not the subject of analysis that we are aware of, we suggest that this should be the topic of additional methodological research. To ensure that the effects shown are not dependent on the ICA thresholds we used, an analysis predicting the grand average spectral slope based on age was also conducted for other correlation thresholds showing an overall similar pattern (see *Figure 4—figure supplement 6*). We further managed to replicate the finding that age has the strongest impact on the spectral slope of the ECG components using a different algorithm to extract aperiodic activity (FOOOF *Donoghue et al., 2020*; see *Figure 4—figure supplements 1–3*). Using FOOOF (*Donoghue et al., 2020*), we also investigated the impact of different slope fitting options (fixed vs. knee model fits) on the aperiodic age relationship (see *Figure 4—figure supplement 3*). The results that we obtained from these analyses using FOOOF offer converging evidence with our main analysis using IRASA. We further replicated our findings on an additional dataset containing resting state recordings (N=655) obtained as part of MEG studies routinely conducted at the University of Salzburg (see *Figure 4—figure supplement 4*). While all these control analyses indicate that the age-related steepening effects occur robustly in the MEG, it is unclear whether they can be exclusively attributed to cardiac activity. Crucially, the topography of the observed steepening effects is present across the scalp and prominent at frontal and temporal sensors around the MEG helmet (albeit also observable at central locations; see MEG$_{ECG Components}$). This topography is suggestive of artifacts that could also be induced by muscle activity (e.g. head/eye movements). We therefore used the subject's head movement information obtained via continuous hpi measurements as a covariate (i.e. 5 coils continuously emitting sinusoidal waves at 293 Hz, 307 Hz, 314 Hz, 321 Hz, and 328 Hz to localize the head position in the scanner). While head movements increased significantly with aging ($\beta_{standardized}$=0.23, *HDI* = [0.18, 0.28], see *Figure 4—figure supplement 8*), it was not sufficient to explain the observed steepening or flattening effects in the spectral slope (see *Figure 4—figure supplement 8*). We further investigated age-related changes to the spectral slope of the vertical and horizontal EOG channels indicating no significant age-related steepening/flattening across the investigated frequency ranges (see *Figure 4—figure supplement 8*). Surprisingly, all these results indicate an age-related steepening in the spectral slope of MEG data both when averaged across sensors and on most individual sensors across two large datasets. This finding is contrary to previous findings showing a flattening of spectral slopes with age in recordings of brain activity (*Voytek et al., 2015*) and cardiac activity (*Beckers et al., 2006*) (see also *Figure 2C*). This discrepancy can potentially be explained by multiple factors including physiologically measured 1 /f noise differently affecting magnetic and electric recording devices (*Dehghani et al., 2010*), preprocessing choices, fitting ranges for the 1 /f slope, electrode selection etc. (see discussion).

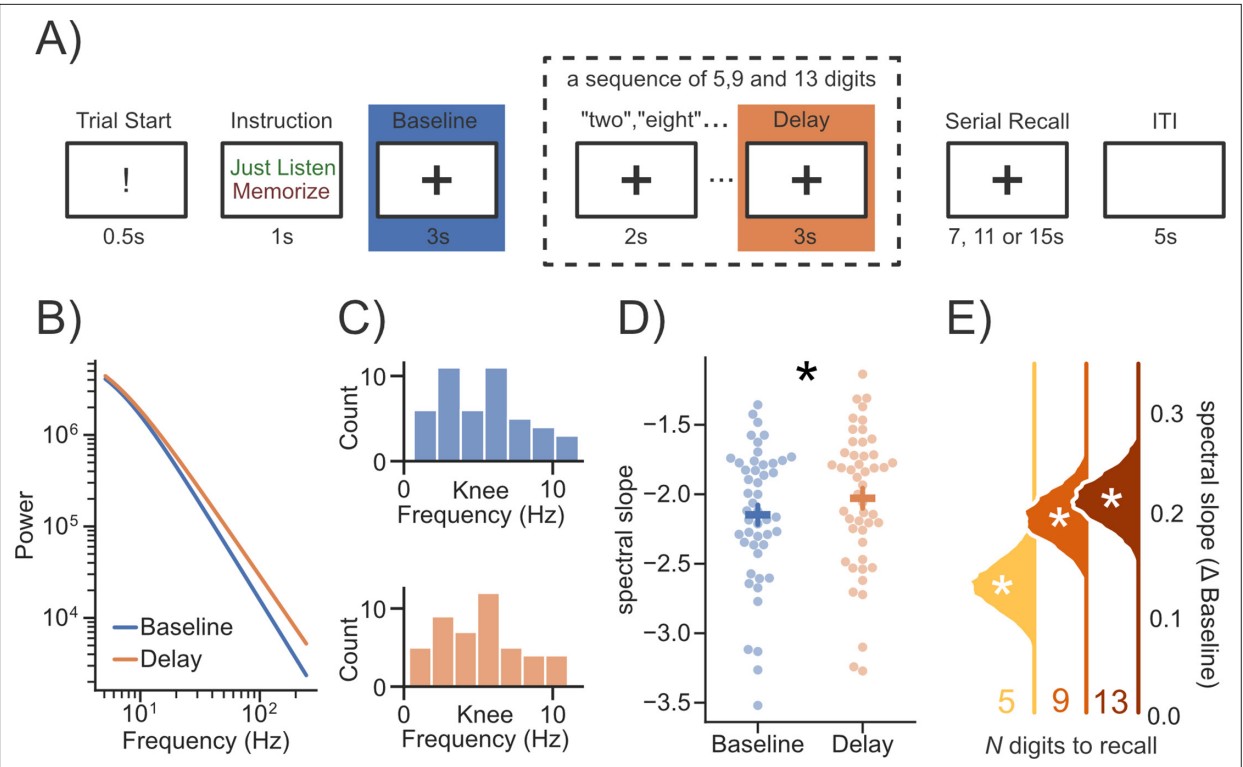

**Figure 5.** Event-related spectral parametrization of working memory in the ECG. (**A**) Subjects were asked to either 'listen' to or 'memorize' a sequence of 5, 9, and 13 digits (adapted from ***Kosachenko et al., 2023***; ***Pavlov et al., 2022***). Spectra in the 'Baseline' period were compared to the 'Delay' period of the 'memorize' condition. (**B**) The averaged evoked difference in the aperiodic spectrum between baseline and delay periods. The spectra were reconstructed from the aperiodic parameters of the spectral fits and plotted as a function of frequency past the average knee frequency (~5 Hz; **C**). The spectral slope of the ECG signal was significantly flatter during the 'Delay' compared to the 'Baseline' period (**D**). (**E**) The flattening of the spectral slope relative to 'Baseline' was strongest in conditions with higher working memory load. Error bars indicate standard errors of the mean. A * indicates a 'significant' effect (see Methods - Statistical Inference). Error bars indicate the 95% confidence interval of the mean.

## Outlook

So far, we have shown that slow physiological changes (e.g. aging) can modulate aperiodic cardiac activity. To further understand the extent by which aperiodic ECG signals are also co-modulated by rapid event-related changes, for example in cognitive tasks, we investigated the ECG recordings of a dataset employing a working memory paradigm (***Kosachenko et al., 2023***; ***Pavlov et al., 2022***; see ***Figure 5A***; for details *Methods - Working Memory Analysis*). Similarly, as in ***Donoghue et al., 2020*** we compared a prestimulus 'Baseline' to a post stimulus 'Delay' period during a working memory task. Interestingly, akin to the EEG results reported by ***Donoghue et al., 2020***, we observed a consistent flattening of the aperiodic slope for cardiac activity in the delay period (see ***Figure 5B,D***; $\beta_{standardized}$=0.23, $HDI$ = [0.16, 0.32]). Furthermore, upon comparing the change of slope relative to the baseline period across different levels of co,gnitive load we noticed that the flattening effect of the slope was modulated by cognitive load (see ***Figure 5E***). The slope flattened the most in the condition with the highest working memory load (13 items; $\beta_{standardized}$=0.42, $HDI$ = [0.33, 0.52]), followed by the high load (9 items; $\beta_{standardized}$=0.39, $HDI$ = [0.29, 0.48]) and the low load condition (5 items; $\beta_{standardized}$=0.25, $HDI$ = [0.15, 0.35]). These results highlight the importance of considering the influence of cardiac activity when investigating changes in aperiodic activity in a state-dependent manner.

## Discussion

Aperiodic processes are ubiquitous in nature (***Gisiger, 2001***). They can be observed not only in physiological recordings but are also found in earthquakes, economics, ecology, epidemics, speech, and music (***Gisiger, 2001***; ***Bak et al., 1987***). In measurements containing multiple aperiodic signals, aperiodic signals might even influence each other (e.g. neural speech tracking ***Chen et al., 2023***; ***Schmidt***

*et al., 2021*). The signals measured using M/EEG reflect a mixture of physiological sources (e.g. cortical, cardiac, myographic, and ocular), each of which exhibits aperiodic and periodic properties. To understand the (a-)periodic signal measured using M/EEG, it is inevitable to understand how these different sources contribute to the (a-)periodic M/EEG signal. This becomes especially important when multiple physiological sources are modulated by the same traits, states, and disorders. Cardiac and cortical recordings of aperiodic electrophysiological signals are related to age (*Voytek et al., 2015*; *Beckers et al., 2006*), cognitive states (e.g. awake vs. sleep *He et al., 2010*; *Penzel et al., 2003*) and disorders such as Parkinson's disease and epilepsy (*Ghinda et al., 2020*; *Mostile et al., 2019*; *Ansakorpi et al., 2002*; *Haapaniemi et al., 2001*). So far, cardiac and cortical activity has mostly been analyzed separately (see *Figure 1C*). In the present study, we investigated whether age-related changes in neural and cardiac aperiodic activity are (in-)dependent. Our results demonstrate that while cardiac activity significantly contributes to age-related changes in aperiodic activity, the extent of this influence varies by frequency range and sensor location—with neural aperiodic activity persisting in some instances. These findings, robust across diverse processing choices and replicable in an independent MEG dataset (see *Control Analyses: Age-related steepening of the spectral slope in the MEG*), may also extend beyond aging. Notably, in a working memory paradigm, the aperiodic spectrum of 'pure' ECG recordings is modulated by working memory load, mirroring previous findings (*Donoghue et al., 2020*) and underscoring the potentially broader impact of cardiac activity on the aperiodic signal recorded using M/EEG.

## Differences in aperiodic activity between magnetic and electric field recordings

Surprisingly, a vast amount of our results using MEG data indicate a steepening of the spectral slope with age. This is contrary to previous findings using mainly EEG/ECoG data (*Voytek et al., 2015*) that commonly show a widespread flattening of the spectral slope with age (*Voytek et al., 2015*). Similarly, we also noticed an age-related flattening on simultaneous ECG recordings (see *Figure 2C*). So do these discrepancies reflect general differences between electric vs. magnetic recordings of physiological activity? Previous research has shown scaling differences in the spectral slope between MEG and EEG recordings (*Dehghani et al., 2010*). These scaling differences are partly widespread (overall flatter sloped spectra in MEG data) and partly regionally specific (steeper sloped spectra at vertex in MEG compared to EEG recordings and vice versa at frontal regions). These observations have been linked to non-resistive properties of tissue (i.e. the propagation of the electric field through tissue is frequency dependent *Dehghani et al., 2010*; *Bedard et al., 2017*). This differently affects the signal recorded using MEG and EEG, as magnetic field recordings are not distorted by the tissue conductivity of the scalp, skull, cerebrospinal fluid, and brain (*Singh, 2014*). An alternative, but not exclusive, hypothesis suggests that even under the assumption of a purely resistive medium (which is unlikely *Bedard et al., 2022*; *Gomes et al., 2016*), frequency scaling differences between MEG and EEG may emerge in relation to the space/frequency structure of the recorded activity (*Bénar et al., 2019*). Under this hypothesis, lower frequencies are suggested to involve synchronous activity in larger patches of cortex, whereas higher frequencies involve synchronous activity in smaller cortical patches. *Bénar et al., 2019* demonstrate that EEG typically integrates activity over larger volumes than MEG, resulting in differently shaped spectra across both recording methods. During aging, both changes in conductive tissue properties (*Mohammed et al., 2017*; *Thomas et al., 2018*) and functional connectivity occur (*Deery et al., 2023*). Hypothetically, an interaction between several factors that differently affect MEG and EEG (e.g. age-related changes in non-resistive properties of tissue and in functional connectivity) may therefore potentially explain differently shaped spectra in MEG compared to EEG recordings. Future research is needed to explore the differences in magnetic and electric field recordings to understand the age-related changes to non-resistive tissue properties alongside age-related changes in functional connectivity. Differences in electric and magnetic field recordings aside, aperiodic activity may not change strictly linearly as we are ageing, and studies looking at younger age groups (e.g. <22; *Tröndle et al., 2022*) may capture different aspects of aging (e.g. brain maturation) than those looking at older subjects (>18 years; our sample). A recent report even shows some first evidence of an interesting putatively non-linear relationship with age in the sensorimotor cortex for resting recordings (*Cross et al., 2024*). Another possible and not mutually exclusive explanation for the age-related steepening could be related to the ECG signal itself. We noticed an age-related

steepening in the spectral slope of the ECG recording in the Cam-Can dataset between ~0.25 and 12 Hz (see *Figure 2C*). Depending on how the power of the aperiodic ECG signal in this low frequency band is reflected on the MEG sensors, this could also bias the spectral slope of the combined MEG/ECG signal. However, an age-related steepening of the ECG was only noted at a frequency range between ~0.25 and 12 Hz making it an unlikely explanation of all the effects we detected in the MEG that span frequency ranges vastly exceeding the 0.25–12 Hz range.

## Influences of preprocessing decisions on age-related changes in aperiodic activity

While differences between magnetic and electric field recordings may explain some observed differences in the widespread effects between electric and magnetic recordings of electrophysiological activity, we still observed flatter sloped spectra with age at a few MEG sensors across several frequency ranges (see *Figure 4C,H,I*). These findings are also in line with previous analyses of MEG data investigating age-related changes in aperiodic activity (*He et al., 2019*; *Thuwal et al., 2021*). The frequency dependence of the flattening/steepening effects (see *Figure 4*) suggests that the slope of the power spectrum can be very sensitive to different preprocessing decisions that may emphasize different aspects of (neuro-)physiological activity. In the case of the MEG signal, this may include the application of SSS algorithms (*Taulu and Simola, 2006*; *Taulu and Kajola, 2005*), different thresholds for ICA component detection (see Figure S7), high and low pass filtering, choices during spectral density estimation (window length/type etc.), different parametrization algorithms (e.g. IRASA vs FOOOF), and selection of frequency ranges for the aperiodic slope estimation. We therefore applied a wide variety of processing settings when analyzing our data. Our results indicate overall steeper sloped spectra with increasing age across datasets and processing options for MEG. These observed steepening effects can be explained by cardiac activity (see *Figure 4H,I*). Cardiac activity, in the form of the ECG, is also captured in EEG recordings (via volume conduction; see *Figure 3A-D*). Our data suggests that the ECG signal is captured equally strong in concurrent MEG and EEG recordings (see *Figure 3A-D*). Furthermore, separating ECG-related components from neural activity using ICA seems to work worse in EEG compared to MEG recordings (see *Figure 3A,B*). Difficulties in removing ECG-related components from EEG signals via ICA might be attributable to various reasons such as the number of available sensors or assumptions related to the non-Gaussianity of the underlying sources. Further understanding of this matter is highly important given that ICA is the most widely used procedure to separate neural from peripheral physiological sources (see *Figure 1E,F*). Additionally, it is worth noting that the effectiveness of an ICA crucially depends on the quality of the extracted components (*Bailey et al., 2023*; *Bailey, 2024*) and even widely suggested settings, for example high-pass filtering at 1 Hz before fitting an ICA may not be universally applicable (see supplementary material of *Bailey, 2024*). Previous work (*Kozhemiako et al., 2022*; *Lendner et al., 2020*) has shown that a linked mastoid reference alone was particularly effective in reducing the impact of ECG-related activity on aperiodic activity measured using EEG. However, it should be considered that depending on the montage, referencing can induce ambiguities to the measured EEG signal. Linked mastoid referencing, for instance, can distort temporal activity (*Feindel et al., 2009*), which is unproblematic in studies focusing on activity on central electrodes (*Kozhemiako et al., 2022*; *Lendner et al., 2020*), but should be considered when focusing on activity from other recording sites (*Yao et al., 2019*).

To better delineate cardiac and neural contributions when investigating aperiodic activity, ECG recordings should become more standard practice in EEG research. Additionally, further method development is needed to better separate cardiac from neural activity in M/EEG recordings.

## (Neuro-)physiological origins of aperiodic activity

Aperiodic activity is present in recordings of different physiological signals, including neural (*He et al., 2010*), cardiac (*Saul et al., 1988*), and muscle activity (*Kozhemiako et al., 2022*). Our study investigated age-related changes in aperiodic activity using MEG and found that these changes vary depending on the frequency range and recording site. Specifically, some changes were found to be uniquely linked to either cardiac or brain activity, while others were present in both signals (*Figure 4H*). Interestingly, some of these shared effects could be attributed to cardiac activity (see *Figure 4I*). However, some effects appeared to be equally influential in explaining age-related changes in both cardiac and brain activity, such that the individual effects disappeared when analyzed jointly (see

*Figure 4I*, upper panel). These findings underscore the complexity of analyzing aperiodic activity, indicating that the aperiodic signal recorded non-invasively originates from multiple physiological sources. These shared effects are particularly interesting, as they suggest that a common mechanism across physiological signals exists that underlies age-related changes of aperiodic activity. In fact, neural and vascular processes are known to interact with each other (*Takarada-Iemata et al., 2019*). Cardiovascular activity, in the form of the ECG, is also captured in M/EEG recordings (via volume conduction; see *Figure 3A-D*). How longitudinal changes in neural and cardiac processes influence age-related changes in aperiodic activity is an exciting research question. This could be investigated in future studies utilizing longitudinal recordings of joint cardiac and neural activity. Understanding the relationship between neural and cardiac aperiodic activity is essential not only for identifying common underlying processes but also for improving our understanding of the individual generative mechanisms of cardiac and neural aperiodic activity.

For example, a current popular hypothesis states that the generative process underlying aperiodic neural activity is mainly attributed to differences in the ratio between excitatory (AMPA) and inhibitory (GABA) currents that influence the slope of the neural power spectrum (*Gao et al., 2017*). Excitatory currents such as AMPA decay faster than inhibitory currents like GABA. This means that flatter power spectra may be indicative of the presence of more excitatory than inhibitory currents and vice versa (steeper sloped power spectra *Gao et al., 2017*). This theory is (in part) based on research showing that GABAergic drugs like propofol (*Gao et al., 2017*) and glutamatergic drugs like ketamine (*Waschke et al., 2021*) modulate the slope of electrophysiologically measured power spectra. However, propofol and ketamine not only influence neural activity, but also influence heart rate variability (a core component of the ECG *Goddard et al., 2021*; *Kanaya et al., 2003*). So, are drug-induced effects on the slope of the power spectrum (measured using surface electrodes) conflated by changes in cardiac activity? Previous work has shown that propofol-induced changes to the spectral slope were still present in EEG recordings after using ICA to reject ECG components from the data (*Lendner et al., 2020*). However, our results suggest that cardiac activity remains in EEG signals even after separating cardiac from neural sources using an ICA with a very sensitive rejection criterion (see *Figure 3A-D*). It is therefore plausible that drug-induced effects on aperiodic "neural" activity can still be conflated by cardiac activity. Future work is needed to see to what extent drug-induced changes in aperiodic neural activity can also be attributed to ECG signals. Similar caveats adhere to other findings of functional modulations in aperiodic signals in cognitive states (e.g. awake vs. sleep *He et al., 2010*; *Penzel et al., 2003*) and disorders like Parkinson's disease and epilepsy (*Ghinda et al., 2020*; *Mostile et al., 2019*; *Ansakorpi et al., 2002*; *Haapaniemi et al., 2001*). This calls for the initiation of - ideally - multicenter coordinated activities aimed to replicate 1 /f aperiodic neural activity effects (e.g. induced by anesthetic drugs) while considering cardiac activity. Another pending research question lies in understanding whether our findings on non-invasive data also translate to data from invasive recordings. Given that cardiac activity is also captured on, for example ECoG (*Kern et al., 2013*), an influence is not unlikely depending on the strength of cardiac activity relative to the measured neural activity.

It is worth noting that, apart from cardiac activity, muscle activity can also be captured in (non-) invasive recordings and may drastically influence measures of the spectral slope (*Fitzgibbon et al., 2016*). To ensure that persistent muscle activity does not bias our results, we used changes in head movement velocity as a control analysis (see *Figure 4—figure supplement 8*). However, it should be noted that this is only a proxy for the presence of persistent muscle activity. Ideally, studies investigating aperiodic activity should also be complemented by measurements of EMG. Whenever such measurements are not available, creative approaches that use the steepness of the spectral slope (or the lack thereof) as an indicator to detect whether or not, for example an independent component is driven by muscle activity are promising (*Fitzgibbon et al., 2016*; *Keil et al., 2022*). However, these approaches may require further validation to determine how well myographic aperiodic thresholds are transferable across the wide variety of different M/EEG devices.

While the present analysis focuses on aperiodic activity, our results might also translate to older findings focusing on "presumably" periodic neural activity in canonical frequency bands (e.g. delta, theta, alpha). Until recently, aperiodic activity was often discarded as noise. Recently developed algorithms have opened up possibilities to separately analyze aperiodic and periodic activity (*Donoghue et al., 2020*; *Wen and Liu, 2016*). This has, for instance, revealed that previously suspected periodic

age-related changes in alpha power may actually be attributable to differences in aperiodic activity (*Donoghue et al., 2020*; *Merkin et al., 2023*). As we have shown that age-related changes in aperiodic activity are linked to cardiac activity, it is possible that our results also translate to previous studies conflating periodic and aperiodic activity. However, whether or not periodic activity (after detection) should be detrended using approaches like FOOOF or IRASA still remains disputed, as incorrectly detrending the data may cause larger errors than not detrending at all (*Brake et al., 2024*).

## Recommendations

Changes in aperiodic activity of peripheral and neural signals are co-modulated by similar traits, states, and disorders. To better disentangle physiological and neural sources of aperiodic activity, we propose the following steps to (1) measure and (2) account for physiological influences.

Measure potential confounding physiological signals explicitly (e.g. ECG) and test whether there is an association between the respective (a-)periodic signal and the feature of interest (e.g. age). In case the feature of interest co-modulates neural and physiological aperiodic activity, it is necessary to account for this. (2) Reduce the influence of physiological signals on neural activity as much as possible. Currently, ICA can be used to at least reduce the impact of cardiac activity (see also *Figure 3*). However, separating physiological from neural sources using an ICA is no guarantee that peripheral physiological activity is fully removed from the cortical signal. Even more sophisticated ICA-based methods that, for example apply wavelet transforms on the ICA components may still not provide a good separation of peripheral physiological and neural activity (*Castellanos and Makarov, 2006*; *Bailey, 2024*). This turns the process of deciding whether or not an ICA component is, for example, either reflective of cardiac or neural activity into a challenging problem. For instance, when we only extract cardiac components using relatively high detection thresholds (e.g. $r > 0.8$), we might end up misclassifying residual cardiac activity as neural. In turn, we can't always be sure that using lower thresholds won't result in misinterpreting parts of the neural effects as cardiac. Both ways of analyzing the data can potentially result in misconceptions. In the present study, we show that our effects are largely consistent across different thresholds (see *Figure 4—figure supplement 6*), but future research should be devoted to developing objective criteria that can be used to make informed decisions when results are inconsistent. Additionally, it might be necessary to invest in the development of new methods to better separate peripheral from neural signals, for instance, combinations of ICA with other methods such as, for example, empirical mode decomposition (see *Wang et al., 2016*; *Teng et al., 2021*). Other promising approaches may potentially involve bipolar referencing for EEG or spatial referencing approaches such as current source density (*Perrin et al., 1987*). How these approaches impact aperiodic activity should be further investigated. For MEG source reconstruction approaches like beamforming may be promising (see e.g. for reduction of tACS artifacts in the MEG *Neuling et al., 2015*). In the case that it is not possible to sufficiently reduce the influence of physiological signals on neural activity, it is necessary to control for this in the statistical model (e.g. by using the confounding physiological aperiodic signal as a covariate).

## Conclusion

The present study, focusing on age-related changes in aperiodic neural and cardiac activity, indicates that the aperiodic signal recorded using surface sensors/electrodes originates from multiple physiological sources. Cardiac and neural age-related changes in aperiodic activity vary depending on the frequency range and recording site. Conflating cardiac and neural contributions to aperiodic activity obstructs our understanding of both neural and cardiac aperiodic processes and should be avoided. These results highlight the need for concurrent recordings of cardiac and neural activity to further increase our understanding of both cortical and cardiac aperiodic activity and its association with age, cognitive states, and disorders.

## Methods

### Sample

The present study builds on resting-state data from several sources. ECG Datasets 1 & 2 were obtained from PhysioNet (*Goldberger et al., 2000*) containing 1121 healthy volunteers between 18 and 92 years of age (divided into 15 age groups for anonymization purposes) (*Schumann and Bär,*

*2021*). The MEG data analyzed in this study was obtained from two different sources: the CamCAN repository (*Figure 4*, *Figure 4—figure supplements 1–8*) and resting-state data routinely recorded at the University of Salzburg (*Figure 3*, *Figure 4—figure supplement 4*). The sample obtained from CamCAN contained 655 healthy volunteers with an age range between 18 and 88 years of age with an average age of 54 and a standard deviation of 18 years. The sample obtained from the University of Salzburg contained 684 healthy volunteers with an age range between 18 and 73 years of age with an average age of 32 and a standard deviation of 14 years. ECG was recorded alongside all MEG recordings. Notably, the age distribution recorded at the University of Salzburg was bimodal (with 423 subjects being below 30, the sample does not reflect an age-diverse population; see *Figure 4—figure supplement 4*). For the ECG data, no specific exclusion criteria for participants were applied. Data from MEG/EEG participants were excluded when no independent heart component ($N_{CamCAN}$ = 18) equal or above the threshold was found ($r$>0.4; see MEG/EEG Processing - pre-processing). Furthermore, when the automated processing procedure resulted in errors, data was considered as missing. This resulted in a total of 1104 ECG recordings from PhysioNet, 627 MEG recordings from CamCAN, and 655 MEG recordings from the University of Salzburg. The CamCAN study was conducted in compliance with the Helsinki Declaration and has been approved by the local ethics committee, Cambridgeshire 2 Research Ethics Committee (reference: 10 /H0308/50). The M/EEG and ECG data obtained at the University of Salzburg were recorded as part of routine resting-state measurements conducted prior to various experimental paradigms, which were approved by the Ethics Committee of the University of Salzburg. Informed consent was obtained from each participant.

## Literature analysis

The literature analysis was performed using the Literature Scanner (LISC) (*Donoghue, 2019*) toolbox and custom written python functions. Briefly, LISC allows for the collection and analysis of abstracts and meta information from scientific articles through a list of search terms. In this manuscript, we used lists of terms for aperiodic activity (e.g. 1 /f, power law, scale-free, etc.), recording devices (ECG, MEG, and EEG) and related association terms (e.g. aging, working memory, etc.) with relevant synonyms. The articles ($N_{Articles}$ = 489) that reference these terms in their abstracts were extracted from the PubMed database. We extracted the proportion of articles related to aperiodic activity and associated with ECG and/or M/EEG (see *Figure 1A*). Furthermore, we displayed the publishing dates of articles in ECG or M/EEG over time (see *Figure 1B*). Additionally, we extracted the article counts of several association terms (e.g. aging) and visualized them separately for ECG and M/EEG. We then extracted the doi's of the articles associated with MEG and EEG separately and sequentially extracted the HTML of the full texts (whenever the texts were accessible). This resulted in 213 articles for EEG and 66 articles for MEG. Afterwards, we extracted search words related to ECG and cardiac activity (with relevant synonyms and word stems; cardio, cardiac, heart, ecg) from each manuscript along the respective word context (400 signs before and 600 after each search term; for each time a search word was mentioned in one of the extracted manuscripts). Manuscripts in which the word stems 'reject' (and related synonyms and word stems; remov, discard, reject) were mentioned in one of the word contexts were temporally marked as 'valid'. The word contexts were further queried for search terms related to common blind source separation artifact rejection approaches such as ICA (*Hyvärinen, 1999*), SVD (*Lagerlund et al., 1997*), SSS (*Taulu and Simola, 2006*), SSP (*Uusitalo and Ilmoniemi, 1997*), and DSS (*de Cheveigné, 2010*). All valid word contexts were then manually inspected by scanning the respective word context to ensure that the removal of 'artifacts' was related specifically to cardiac and not for example ocular activity or the rejection of artifacts in general (without specifying which 'artifactual' source was rejected in which case the manuscript t was marked as invalid). This means that the results of our literature analysis likely present a lower bound for the rejection of cardiac activity in the M/EEG literature investigating aperiodic activity. Finally, we visualized the proportion of articles in relation to the respective search words (see *Figure 1D,E*). Furthermore, we arbitrarily selected 60 articles investigating aperiodic activity and visualized the investigated frequency ranges alongside their respective upper and lower bounds (*Figure 1F-H*).

## Statistical inference

To investigate the relationship between age and aperiodic activity recorded using MEG, we used Bayesian generalized linear models (GLMs) either built directly in PyMC (a Python package for

probabilistic programming *Salvatier et al., 2016*) or in Bambi (a high-level interface to PyMC *Capretto, 2022*). Decisions for either Bambi or PyMC were made based on the accessibility of appropriate statistical families for the respective dependent variables in a GLM. Priors were chosen to be weakly informative (*Westfall, 2017*) (exact prior specifications for each model can be obtained from the code in the corresponding authors' GitHub repository; see Data availability). Results were considered statistically significant if 94% of the highest (probability) density interval (HDI) of the posterior for a given standardized β-coefficient or (partial) correlation coefficient was not overlapping with a region of practical equivalence between –0.1 and 0.1 as suggested by *Kruschke, 2018* based on negligible effect sizes according to *Cohen, 1988*. Furthermore, it was ensured that for all models there were no divergent transitions ($R_{hat}$ <1.05 for all relevant parameters) and an effective sample size >400 (an exhaustive summary of Bayesian model diagnostics can be found in *Vehtari et al., 2021*).

## MEG/EEG processing

### MEG/EEG processing - data acquisition

MEG data (CamCAN; *Figure 4*) was recorded at the University of Cambridge, using a 306 VectorView system (Elekta Neuromag, Helsinki). MEG data (Salzburg; *Figure 3*, *Figure 4—figure supplement 4*) was recorded at the University of Salzburg, using a 306 channel TRIUX system (MEGIN Oy, Helsinki). Both systems are equipped with 102 magnetometers and 204 planar gradiometers and positioned in magnetically shielded rooms. In order to facilitate offline artifact correction, electrooculogram (VEOG, HEOG) as well as ECG was measured continuously in both recording sites. In a subset of the Salzburg recordings, EEG was measured additionally (see *Figure 3A-D*) using a 32-channel system provided by the MEG manufacturer. Data recorded at Cambridge was online filtered at 0.03–330 Hz, whereas data recorded at Salzburg was online filtered at 0.1–333 Hz with a 1000 Hz sampling rate at both recording sites. Five Head-Position Indicator coils were used to measure the position of the head. All data used in the present study contains passive resting-state measurements lasting about ~8 min (Cambridge) and ~5 min (Salzburg). Further data processing at both recording sites was conducted similarly and will therefore be reported together.

### MEG/EEG processing - pre-processing

Initially, an SSS (*Taulu and Simola, 2006*; *Taulu and Kajola, 2005*) algorithm was used to find and repair bad channels (implemented in MNE-Python *version 1.2 Gramfort et al., 2013*). The data was further processed by either not applying further SSS cleaning (main manuscript) or by applying an SSS algorithm for additional data cleaning (*Figure 4—figure supplement 5* & *Figure 4—figure supplement 7*; implemented in MNE-Python *version 1.2 Gramfort et al., 2013*). The data were afterwards high-pass filtered at 0.1 Hz using a finite impulse response (FIR) filter (Hamming window). EEG data was re-referenced to the common average (common practice in the field *Yao et al., 2019*). For extracting physiological 'artifacts' from the data, 50 independent components were calculated using the *fastica* algorithm (*Hyvärinen, 1999*; implemented in MNE-Python *version 1.2*; with the parallel/symmetric setting; note: 50 components were selected for MEG for computational reasons for the analysis of EEG data no threshold was applied). As ICA is sensitive to low-frequency drifts, independent components were calculated on a copy of the data high-pass filtered at 1 Hz. Components related to cardiac and ocular activity were determined via correlation with concurrent ECG, VEOG, and HEOG recordings. A threshold of $r$>0.4 was applied to detect the ECG components and a threshold $r$>0.8 for detecting EOG components in the data. The more sensitive threshold used for ECG component detection was decided upon based on the strong presence of ECG signals in resting state M/EEG recordings (see *Figure 2E-H*). The computed ICA was then applied to the original data, either rejecting all components apart from those related to the ECG, or rejecting only EOG-related components, or ECG and EOG-related components. This resulted in three conditions $MEG_{ECG\ not\ rejected}$, $MEG_{ECG\ rejected}$, and $MEG_{ECG\ component}$. The data were then split into 2 s epochs and residual artifacts were determined using an adaptive and automatic artifact detection method (the 'Riemannian Potato' implemented in pyriemann *Barachant et al., 2013*). Epochs were rejected when the covariance matrix of an epoch differed by >2.5 standard deviations from a centroid covariance matrix.

## MEG/EEG processing - temporal response functions

To estimate the extent at which ECG activity is captured by MEG/EEG recordings, we calculated temporal response functions (TRFs). In brief, the assumption behind a TRF is that a dependent variable is the result of one (or several) predictor variables. This approach can be used to model a time-series as a linear function of one (or several) time series and can be formulated (for a single predictor *Brodbeck et al., 2021*) as:

$$y_t = \sum_{\tau=\tau_{min}}^{\tau_{max}} h_\tau x_{t-\tau}$$

where h represents the TRF, sometimes described as filter kernel, and $\tau$ represents potential delays between y and x (for an extensive explanation of the algorithm used herein see *Brodbeck et al., 2021*). Typically, this approach is used to estimate spectro-temporal receptive fields, for example in response to auditory stimuli (*David et al., 2007*), where the interpretation of a TRF follows that of an ERP/F in response to a continuous input signal (*Lalor et al., 2006*). This model can be used as a forward or encoding model to predict the brain response from a stimulus representation or as a backward or decoding model to reconstruct a stimulus representation from brain recordings. We used this approach here to detect whether ECG and M/EEG influence each other. Concurrent recordings of MEG and EEG data alongside ECG were only available for a subset of subjects (N=20). We therefore selected those subjects from the subjects data pool of the Salzburg sample to perform the TRF analysis. Forward and Backward models were calculated between the M/EEG signal and the signal recorded using ECG. This was done to see if signals measured using M/EEG and ECG follow/precede each other (non zero-time lags; e.g. via interoception see *Schulz, 2016*) or if the signals are instantaneously related (zero-time-lags; therefore likely corresponding to the same underlying signal source). As computations of TRFs are memory and time extensive, the MEG, EEG, and ECG data were additionally low-pass filtered at 45 Hz (to avoid aliasing) and downsampled to 100 Hz before the analysis (a common practice when computing TRFs; see *Brodbeck et al., 2021*). For the computation of the TRFs, the ECG and M/EEG data were normalized (z-scored), and an integration window from −250 to 250ms with a kernel basis of 50ms Hamming windows was defined. To prevent overfitting, model parameters were adjusted using a four-fold nested cross-validation (two training folds, one validation fold, and one test fold), each partition served as a test set once. The accuracy of the model was assessed by calculating the Pearson correlation coefficient between the respective predicted and measured time series. We calculated the same model in the forward direction (encoding model; i.e. predicting M/EEG data in a multivariate model from the ECG signal) and backward direction (decoding model; i.e. predicting the ECG signal using all M/EEG channels as predictors). The accuracy of the decoding model was used in *Figure 3C,D* to assess how well the ECG time series was decodable from M/EEG data. For the respective encoding model, we visualized the non-normalized encoding weights (see *Figure 3A,B*). The TRF time courses visualized in *Figure 3A* were obtained by computing a principal component analysis (PCA *Pearson, 1901*) within subject and across all channels, whereas the distribution of peaks visualized in *Figure 3B* was obtained by calculating the root mean square across channels and extracting the maximum value.

## MEG/EEG processing - spectral analysis

Power spectra were computed using Welch's method (*Welch, 1967*) between 0.1 and 145 Hz (0.5 Hz resolution). Aperiodic activity was extracted using the IRASA method (*Wen and Liu, 2016*) implemented in the YASA package (*Vallat and Walker, 2021*). Furthermore, in addition to the main model fit between 0.1 and 145 Hz, additional slopes were fitted to the aperiodic spectrum in 5 Hz steps starting from 45 Hz as upper frequency limit and in 1 Hz steps from 0.5 Hz to 10 Hz as lower frequency limit. Additionally, to investigate the robustness of our result the spectral parameterization algorithm (implemented in *FOOOF* version 1.0.0 *Donoghue et al., 2020*) was used to parametrize raw power spectra. Power spectra were parameterized across frequency ranges of 0.5–145 Hz. FOOOF models were fit using the following settings: peak width limits: [1 − 6]; max number of peaks: 2; minimum peak height: 0.0; peak threshold: 2.0; aperiodic mode: 'fixed'. Goodness of fit metrics for both IRASA and FOOOF can be found in *Figure 4—figure supplement 1*.

**ECG processing**

The ECG data recorded as part of the MEG recordings were processed alongside the MEG data. Therefore, the pre-processing and spectral analysis settings from the section 'MEG/EEG Processing' also apply to the ECG aspect of datasets 3 and 4 (see *Figure 2*). Below, ECG processing for the data obtained from PhysioNet is described.

## ECG processing - data acquisition

The ECG data obtained from PhysioNet were acquired at the Jena university hospital (*Schumann and Bär, 2021*). The study was approved by the ethics committee of the Medical Faculty of the Friedrich Schiller University Jena. All research was performed in accordance with relevant guidelines and regulations. The informed written consent was obtained from all subjects. ECG data were recorded at a sampling rate of 1000 Hz using one of two different recording devices. Either an MP150 (ECG100C, BIOPAC systems inc, Golata, CA, USA) or a Task Force Monitor system (CNSystems Medizintechnik GmbH, Graz, AUT). More detailed information about the ECG recordings can be obtained from physionet.org. The data were further analyzed using spectral analysis.

## ECG processing - spectral analysis

Power spectra were computed using Welch's method (*Welch, 1967*) implemented in *neurodsp* (*Cole et al., 2019*) between 0.25 and 145 Hz (0.1 Hz resolution). The spectral parameterization algorithm (implemented in *FOOOF* version 1.0.0 *Donoghue et al., 2020*) and IRASA (implemented in the YASA package *Vallat and Walker, 2021*) were then used to parametrize the power spectra. Power spectra were parameterized across a frequency range of 0.25–145 Hz using the same settings specified in 'MEG/EEG Processing - Spectral analysis'. However, the aperiodic mode was set to 'knee' given that the power spectra (on average) showed a clear 'knee' in log-log coordinates (see *Figure 2—figure supplement 1*). Additionally, we fitted several slopes to the aperiodic spectrum split in a lower range (0.25–20 Hz) and a higher range (10–145 Hz). The split in low- and high-frequency range was performed to avoid spectral knees at ~15 Hz in the center of the slope fitting range.

## ECG processing - heart rate variability analysis

Heart rate variability (HRV) was computed using the NeuroKit2 toolbox, a high level tool for the analysis of physiological signals. First, the raw ECG data were preprocessed by high-pass filtering the signal at 0.5 Hz using an infinite impulse response (IIR) Butterworth filter (order = 5) and by smoothing the signal with a moving average kernel with the width of one period of 50 Hz to remove the powerline noise (default settings of neurokit.ecg.ecg_clean). Afterward, QRS complexes were detected based on the steepness of the absolute gradient of the ECG signal. Subsequently, R-Peaks were detected as local maxima in the QRS complexes (default settings of neurokit.ecg.ecg_peaks; see *Brammer, 2020* for a validation of the algorithm). From the cleaned R-R intervals, 90 HRV indices were derived, encompassing time-domain, frequency-domain, and non-linear measures. Time-domain indices included standard metrics such as the mean and standard deviation of the normalized R-R intervals, the root mean square of successive differences, and other statistical descriptors of interbeat interval variability. Frequency-domain analyses were performed using power spectral density estimation, yielding, for instance, low-frequency (0.04–0.15 Hz) and high-frequency (0.15–0.4 Hz) power components. Additionally, non-linear dynamics were characterized through measures such as sample entropy, detrended fluctuation analysis, and various Poincaré plot descriptors. All these measures were then related to the slopes of the low- frequency (0.25–20 Hz) and high-frequency (10–145 Hz) aperiodic spectrum of the raw ECG.

**Working memory analysis**

## Sample

As an outlook to which extent the present findings, focused on aging, may translate to studies investigating changes in 'cortical' aperiodic in other settings (e.g. in a state dependent) we analyzed data from a working memory paradigm (*Kosachenko et al., 2023*; *Pavlov et al., 2022*; see *Figure 5A* for an overview of the experimental paradigm). The original study included 86 healthy volunteers. Due to technical difficulties, ECG and EEG recordings were only available for a subset of subjects.

In the present study, we analyzed the data of 48 subjects. The age range of the investigated sample was between 18 and 24 years with an average age of 20 years. Informed consent was obtained from each participant and the experimental protocol was approved by the Ural Federal University ethics committee.

## Digit span task

Each trial began with an exclamation mark for 0.5 s along with a recorded voice command 'begin' – indicating the start of the trial. The exclamation mark was followed by an instruction to either memorize the subsequent digits in the correct order (memory condition) or to just listen to the digits without attempting to memorize them (control condition). The instruction was followed by a 3-s 'Baseline' period. Then either 5, 9, or 13 digits were presented auditorily with an interstimulus interval of 2 s. The digits were presented with a female voice in Russian. Each of the digits from 0 to 9 was used, and the mean duration of each digit was 664ms (min: 462ms, max: 813ms). The last digit in the sequence was followed by a 3 s 'Delay' period. During the baseline, encoding, and 'Delay' period, participants were fixating a cross (1.2 cm in diameter) on the screen. In the memory condition, the participants were asked to recall each digit out loud in the correct order starting from the first one (i.e. serial recall). The retrieval was recorded by a computer microphone controlled by PsychoPy (*Peirce, 2007*). The participants had 7, 11, and 15 seconds for 5, 9, and 13 digit sequences, respectively, to recall the digits. The retrieval was followed by an inter-trial interval of 5 s. In the control condition (passive listening), presentation of the digits and the 'Delay' period was followed immediately by an inter-trial interval of the same duration. There were 9 blocks in total with 54 passive listening and 108 memory trials overall. Each block consisted of 3 control (one of each load) followed by 12 memory (4 trials on each level of load, in random order) followed again by 3 control trials. Before the main working memory task, each participant completed 6 practice trials (3 passive listening and 3 memory trials).

## Data acquisition

ECG was recorded from one channel with the active electrode placed on the right wrist and the reference electrode on the left wrist, and the ground on the left inner forearm at 3 cm distally from the elbow.

## Data analysis

The continuous ECG data was high-pass filtered at 0.1 Hz using a finite impulse response (FIR) filter (Hamming window) and downsampled to 500 Hz after applying an anti-aliasing filter. The downsampled data was then split into 3 s epochs separately for the 'Baseline' and 'Delay' periods, as well as the amount of presented digits (5, 9, or 13). The epoched data was further analyzed either by calculating power spectra over the data split in the 'Baseline' and 'Delay' condition irrespective of the amount of presented digits (*Figure 5B-D*) or by retaining the information about the amount of presented digits (*Figure 5E*). Power spectra were computed using Welch's method (*Welch, 1967*) between 0.1 and 245 Hz (0.333 Hz resolution). The spectral parameterization algorithm (implemented in *FOOOF* version 1.0.0 *Donoghue et al., 2020*) was then used to parametrize the power spectra. Power spectra were parameterized across a frequency range of 0.1–245 Hz using the same settings specified in "MEG/EEG Processing - Spectral analysis". However, the aperiodic mode was set to 'knee' given that the power spectra (on average) showed a clear 'knee' in log-log coordinates.

## Statistical analysis

To investigate the relationship between working memory load and aperiodic activity recorded via ECG, we implemented Bayesian linear mixed-effect models in Bambi (*Capretto, 2022*) using the following formulas according to the Wilkinson notation (*Wilkinson and Rogers, 1973*) with the 'Baseline' condition set as *Intercept* in the following models.

$$spectral\ slope\ \sim\ 1\ +\ Delay\ +\ \big(1|subject\ id\big)$$ (Figure 5D)

$$spectral\ slope\ \sim\ 1\ +\ N\ Digits\ to\ recall\ +\ \big(1|subject\ id\big)$$ (Figure 5E)

## Data visualization

Individual plots were generated in Python using Matplotlib, Seaborn, and MNE-Python. Plots were then arranged as cohesive figures with Affinity Designer (https://affinity.serif.com/en-us/designer/).

## Acknowledgements

The authors thank Tzvetan Popov, Freek van Ede, Rik Henson, and Kamen Tsvetanov for helpful comments and discussions and Thomas Hartmann for a code review. This research was funded in whole or in part by the Austrian Science Fund (FWF) [10.55776/W1233]. For open access purposes, the author has applied a CC BY public copyright license to any author-accepted manuscript version arising from this submission. The authors declare no conflicts of interest.

## Additional information

### Funding

| Funder | Grant reference number | Author |
| --- | --- | --- |
| Austrian Science Fund | 10.55776/W1233 | Sarah K Danböck |

The funders had no role in study design, data collection and interpretation, or the decision to submit the work for publication.

### Author contributions

Fabian Schmidt, Conceptualization, Software, Formal analysis, Investigation, Visualization, Methodology, Writing - original draft, Writing – review and editing; Sarah K Danböck, Formal analysis, Writing – review and editing; Eugen Trinka, Dominic P Klein, Writing – review and editing; Gianpaolo Demarchi, Supervision, Methodology, Writing – review and editing; Nathan Weisz, Conceptualization, Supervision, Writing – review and editing

### Author ORCIDs

Fabian Schmidt ⓘ https://orcid.org/0000-0002-9839-1614
Sarah K Danböck ⓘ https://orcid.org/0000-0001-9989-1146
Gianpaolo Demarchi ⓘ https://orcid.org/0000-0002-7597-9298
Nathan Weisz ⓘ https://orcid.org/0000-0001-7816-0037

### Ethics

Human subjects: The ECG data obtained from PhysioNet were acquired at the Jena university hospital. The study was approved by the ethics committee of the Medical Faculty of the Friedrich Schiller University Jena. The informed written consent was obtained from all subjects. The experimental protocol for the ECG data used in the Working Memory Analysis was approved by the Ural Federal University ethics committee. Informed consent was obtained from each participant. The Cam-CAN study was conducted in compliance with the Helsinki Declaration, and has been approved by the local ethics committee, Cambridgeshire 2 Research Ethics Committee (reference: 10/H0308/50). Informed consent was given by each participant. The MEG/EEG data obtained at the University of Salzburg were recorded as part of routine resting-state measurements conducted prior to various experimental paradigms, which were approved by the Ethics Committee of the University of Salzburg. Informed consent was obtained from each participant.

Reviewer #1 (Public review): https://doi.org/10.7554/eLife.100605.3.sa1
Reviewer #2 (Public review): https://doi.org/10.7554/eLife.100605.3.sa2
Reviewer #3 (Public review): https://doi.org/10.7554/eLife.100605.3.sa3
Author response https://doi.org/10.7554/eLife.100605.3.sa4

# Additional files

## Supplementary files
MDAR checklist

## Data availability
The M/EEG and ECG data analyzed in the main manuscript are mostly obtained from open data sources. The ECG data (Dataset 1 & 2; Figure 2) were obtained from https://physionet.org/content/autonomic-aging-cardiovascular/1.0.0/. The data underlying Figure 3 and Figure 4-figure supplement 4 are available at https://osf.io/jcekr/. The data for the MEG analysis in Figure 4 (and Figure 4-figure supplement 1, 2, 3, 5, 6, 7 & 8) are obtained from https://cam-can.mrc-cbu.cam.ac.uk/functional/. The ECG data for the working memory analysis was obtained from https://openneuro.org/datasets/ds003838/versions/1.0.5 (Figure 5). All code used for the analysis is publicly available on GitHub at: https://github.com/schmidtfa/cardiac_1_f (copy archived at *Schmidt and Hartmann, 2023*) & https://github.com/schmidtfa/ecg_1f_memory, (copy archived at *Schmidt, 2023*).

The following dataset was generated:

| Author(s) | Year | Dataset title | Dataset URL | Database and Identifier |
|---|---|---|---|---|
| Schmidt F | 2025 | cardiac_1_f_data | https://osf.io/jcekr/ | Open Science Framework, jcekr |

The following previously published datasets were used:

| Author(s) | Year | Dataset title | Dataset URL | Database and Identifier |
|---|---|---|---|---|
| Yuri GP, Kasanov D, Kosachenko AI, Kotyusov AI | 2022 | EEG, pupillometry, ECG and photoplethysmography, and behavioral data in the digit span task and rest | https://openneuro.org/datasets/ds003838/versions/1.0.5 | OpenNeuro, 10.18112/openneuro.ds003838.v1.0.5 |
| Shafto MA, Tyler LK, Dixon M, Taylor JR, Rowe JB, Cusack R, Calder AJ, Marslen-Wilson WD, Duncan J, Dalgleish T, Henson RN, Brayne C, Matthews FE, Cam-CAN | 2014 | Cambridge Centre for Ageing and Neuroscience (CamCAN) | https://opendata.mrc-cbu.cam.ac.uk/projects/camcan/ | Cam-CAN, camcan |

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
