## [Editor Report · eLife Assessment]

Examination of (a)periodic brain activity has gained particular interest in the last few years in the neuroscience fields relating to cognition, disorders, and brain states. Using large EEG/MEG datasets from younger and older adults, the current study provides **compelling** evidence that age-related differences in aperiodic EEG/MEG signals can be driven by cardiac rather than brain activity. Their findings have **important** implications for all future research that aims to assess aperiodic neural activity, suggesting control for the influence of cardiac signals is essential.

---

## [Referee Report · Reviewer #1 (Public review)]

Summary:

The present study addresses whether physiological signals influence aperiodic brain activity with a focus on age-related changes. The authors report age effects on aperiodic cardiac activity derived from ECG in low and high-frequency ranges in roughly 2300 participants from four different sites. Slopes of the ECGs were associated with common heart variability measures, which, according to the authors, shows that ECG, even at higher frequencies, conveys meaningful information. Using temporal response functions on concurrent ECG and M/EEG time series, the authors demonstrate that cardiac activity is instantaneously reflected in neural recordings, even after applying ICA analysis to remove cardiac activity. This was more strongly the case for EEG than MEG data. Finally, spectral parameterization was done in large-scale resting-state MEG and ECG data in individuals between 18 and 88 years, and age effects were tested. A steepening of spectral slopes with age was observed, particularly for ECG and, to a lesser extent, in cleaned MEG data in most frequency ranges and sensors investigated. The authors conclude that commonly observed age effects on neural aperiodic activity can mainly be explained by cardiac activity.

Strengths:

Compared to previous investigations, the authors demonstrate effects of aging on the spectral slope in the currently largest MEG dataset with equal age distribution available. Their efforts of replicating observed effects in another large MEG dataset and considering potential confounding by ocular activity, head movements, or preprocessing methods are commendable and highly valuable to the community. This study also employs a wide range of fitting ranges and two commonly used algorithms for spectral parameterization of neural and cardiac activity, hence providing a comprehensive overview of the impact of methodological choices. The authors discuss their findings in-depth and give recommendations for the separation of physiological and neural sources of aperiodic activity.

Weaknesses:

While the study's aim is well-motivated and analyses rigorously conducted, it remains vague what is reflected in the ECG at higher frequency ranges that contributed to the confounding of the age effects in the neural data. However, the authors address this issue in their discussion.

---

## [Referee Report · Reviewer #2 (Public review)]

As remains obvious from my previous reviews, I still consider this to be an important paper and that is final and publishable in its current state.

In that previous review, I revealed my identity to help reassure the authors that I was doing my best to remain unbiased because I work in this area and some of the authors' results directly impact my prior research. I was genuinely excited to see the earlier preprint version of this paper when it first appeared. I get a lot of joy out of trying to - collectively, as a field - really understand the nature of our data, and I continue to commend the authors here for pushing at the sources of aperiodic activity!

In their manuscript, Schmidt and colleagues provide a very compelling, convincing, thorough, and measured set of analyses. Previously I recommended that the push even further, and they added the current Figure 5 analysis of event-related changes in the ECG during working memory. In my opinion this result practically warrants a separate paper its own!

The literature analysis is very clever, and expanded upon from any other prior version I've seen.

In my previous review, the broadest, most high-level comment I wanted to make was that authors are correct. We (in my lab) have tried to be measured in our approach to talking about aperiodic analyses - including adopting measuring ECG when possible now - because there are so many sources of aperiodic activity: neural, ECG, respiration, skin conductance, muscle activity, electrode impedances, room noise, electronics noise, etc. The authors discuss this all very clearly, and I commend them on that. We, as a field, should move more toward a model where we can account for all of those sources of noise together. (This was less of an action item, and more of an inclusion of a comment for the record.)

I also very much appreciate the authors' excellent commentary regarding the physiological effects that pharmacological challenges such as propofol and ketamine also have on non-neural (autonomic) functions such as ECG. Previously I also asked them to discuss the possibility that, while their manuscript focuses on aperiodic activity, it is possible that the wealth of literature regarding age-related changes in "oscillatory" activity might be driven partly by age-related changes in neural (or non-neural, ECG-related) changes in aperiodic activity. They have included a nice discussion on this, and I'm excited about the possibilities for cognitive neuroscience as we move more in this direction.

Finally, I previously asked for recommendations on how to proceed. The authors convinced me that we should care about how the ECG might impact our field potential measures, but how do I, as a relative novice, proceed. They now include three strong recommendations at the end of their manuscript that I find to be very helpful.

As was obvious from previous review, I consider this to be an important and impactful cautionary report, that is incredibly well supported by multiple thorough analyses. The authors have done an excellent job responding to all my previous comments and concerns and, in my estimation, those of the previous reviewers as well.

---

## [Referee Report · Reviewer #3 (Public review)]

Summary:

Schmidt et al., aimed to provide an extremely comprehensive demonstration of the influence cardiac electromagnetic fields have on the relationship between age and the aperiodic slope measured from electroencephalographic (EEG) and magnetoencephalographic (MEG) data.

Strengths:

Schmidt et al., used a multiverse approach to show that the cardiac influence on this relationship is considerable, by testing a wide range of different analysis parameters (including extensive testing of different frequency ranges assessed to determine the aperiodic fit), algorithms (including different artifact reduction approaches and different aperiodic fitting algorithms), and multiple large datasets to provide conclusions that are robust to the vast majority of potential experimental variations.

The study showed that across these different analytical variations, the cardiac contribution to aperiodic activity measured using EEG and MEG is considerable, and likely influences the relationship between aperiodic activity and age to a greater extent than the influence of neural activity.

Their findings have significant implications for all future research that aims to assess aperiodic neural activity, suggesting control for the influence of cardiac fields is essential.

Weaknesses:

The authors have addressed the weaknesses of their study in their manuscript. Most alternative explanations for their results have been explored to ensure their conclusions are robust and are not explained by unexplored confounds. Minor potential weaknesses are:

(1) The number of electrodes used in the EEG analyses was on the lower side, and as such, the results do not confirm that the influence of ECG on the 1/f activity in the EEG is high even for higher density EEG montages where ICA may provide better performance at removing cardiac components (as noted by the authors). Having noted this potential weakness, I doubt the effects of cardiac activity can be completely mitigated with current methods, even in higher-density EEG montages.

(2) Head movements were used as a proxy for muscle activity. However, this may imperfectly address the potential influence of muscle activity on the slope in the EEG activity. As such, remaining muscle artifacts may have affected some of the results, particularly those that included high frequency ranges in the aperiodic estimate. Perhaps if muscle activity were left in the EEG data, it could have disrupted the ability to detect a relationship between age and 1/f slope in a way that didn't disrupt the same relationship in the cardiac data. However, I doubt this would reverse the overall conclusions given the number of converging results, including in lower frequency bands. The authors also note this potential weakness and suggest how future research might address it.

---

## [Author Response]

The following is the authors’ response to the original reviews

**eLife Assessment**
Examination of (a)periodic brain activity has gained particular interest in the last few years in the neuroscience fields relating to cognition, disorders, and brain states. Using large EEG/MEG datasets from younger and older adults, the current study provides compelling evidence that age-related differences in aperiodic EEG/MEG signals can be driven by cardiac rather than brain activity. Their findings have important implications for all future research that aims to assess aperiodic neural activity, suggesting control for the influence of cardiac signals is essential.

We want to thank the editors for their assessment of our work and highlighting its importance for the understanding of aperiodic neural activity. Additionally, we want to thank the three present and four former reviewers (at a different journal) whose comments and ideas were critical in shaping this manuscript to its current form. We hope that this paper opens up many more questions that will guide us - as a field - to an improved understanding of how “cortical” and “cardiac” changes in aperiodic activity are linked and want to invite readers to engage with our work through eLife’s comment function.

**Public Reviews:**

**Reviewer #1 (Public review):**
Summary:The present study addresses whether physiological signals influence aperiodic brain activity with a focus on age-related changes. The authors report age effects on aperiodic cardiac activity derived from ECG in low and high-frequency ranges in roughly 2300 participants from four different sites. Slopes of the ECGs were associated with common heart variability measures, which, according to the authors, shows that ECG, even at higher frequencies, conveys meaningful information. Using temporal response functions on concurrent ECG and M/EEG time series, the authors demonstrate that cardiac activity is instantaneously reflected in neural recordings, even after applying ICA analysis to remove cardiac activity. This was more strongly the case for EEG than MEG data. Finally, spectral parameterization was done in large-scale resting-state MEG and ECG data in individuals between 18 and 88 years, and age effects were tested. A steepening of spectral slopes with age was observed particularly for ECG and, to a lesser extent, in cleaned MEG data in most frequency ranges and sensors investigated. The authors conclude that commonly observed age effects on neural aperiodic activity can mainly be explained by cardiac activity.Strengths:Compared to previous investigations, the authors demonstrate the effects of aging on the spectral slope in the currently largest MEG dataset with equal age distribution available. Their efforts of replicating observed effects in another large MEG dataset and considering potential confounding by ocular activity, head movements, or preprocessing methods are commendable and valuable to the community. This study also employs a wide range of fitting ranges and two commonly used algorithms for spectral parameterization of neural and cardiac activity, hence providing a comprehensive overview of the impact of methodological choices. Based on their findings, the authors give recommendations for the separation of physiological and neural sources of aperiodic activity.Weaknesses:While the aim of the study is well-motivated and analyses rigorously conducted, the overall structure of the manuscript, as it stands now, is partially misleading. Some of the described results are not well-embedded and lack discussion.

We want to thank the reviewer for their comments focussed on improving the overall structure of the manuscript. We agree with their suggestions that some results could be more clearly contextualized and restructured the manuscript accordingly.

**Reviewer #2 (Public review):**
I previously reviewed this important and timely manuscript at a previous journal where, after two rounds of review, I recommended publication. Because eLife practices an open reviewing format, I will recapitulate some of my previous comments here, for the scientific record.In that previous review, I revealed my identity to help reassure the authors that I was doing my best to remain unbiased because I work in this area and some of the authors' results directly impact my prior research. I was genuinely excited to see the earlier preprint version of this paper when it first appeared. I get a lot of joy out of trying to - collectively, as a field - really understand the nature of our data, and I continue to commend the authors here for pushing at the sources of aperiodic activity!In their manuscript, Schmidt and colleagues provide a very compelling, convincing, thorough, and measured set of analyses. Previously I recommended that the push even further, and they added the current Figure 5 analysis of event-related changes in the ECG during working memory. In my opinion this result practically warrants a separate paper its own!The literature analysis is very clever, and expanded upon from any other prior version I've seen.In my previous review, the broadest, most high-level comment I wanted to make was that authors are correct. We (in my lab) have tried to be measured in our approach to talking about aperiodic analyses - including adopting measuring ECG when possible now - because there are so many sources of aperiodic activity: neural, ECG, respiration, skin conductance, muscle activity, electrode impedances, room noise, electronics noise, etc. The authors discuss this all very clearly, and I commend them on that. We, as a field, should move more toward a model where we can account for all of those sources of noise together. (This was less of an action item, and more of an inclusion of a comment for the record.)I also very much appreciate the authors' excellent commentary regarding the physiological effects that pharmacological challenges such as propofol and ketamine also have on non-neural (autonomic) functions such as ECG. Previously I also asked them to discuss the possibility that, while their manuscript focuses on aperiodic activity, it is possible that the wealth of literature regarding age-related changes in "oscillatory" activity might be driven partly by age-related changes in neural (or non-neural, ECG-related) changes in aperiodic activity. They have included a nice discussion on this, and I'm excited about the possibilities for cognitive neuroscience as we move more in this direction.Finally, I previously asked for recommendations on how to proceed. The authors convinced me that we should care about how the ECG might impact our field potential measures, but how do I, as a relative novice, proceed. They now include three strong recommendations at the end of their manuscript that I find to be very helpful.As was obvious from previous review, I consider this to be an important and impactful cautionary report, that is incredibly well supported by multiple thorough analyses. The authors have done an excellent job responding to all my previous comments and concerns and, in my estimation, those of the previous reviewers as well.

We want to thank the reviewer for agreeing to review our manuscript again and for recapitulating on their previous comments and the progress the manuscript has made over the course of the last ~2 years. The reviewer's comments have been essential in shaping the manuscript into its current form. Their feedback has made the review process truly feel like a collaborative effort, focused on strengthening the manuscript and refining its conclusions and resulting recommendations.

**Reviewer #3 (Public review):**
Summary:Schmidt et al., aimed to provide an extremely comprehensive demonstration of the influence cardiac electromagnetic fields have on the relationship between age and the aperiodic slope measured from electroencephalographic (EEG) and magnetoencephalographic (MEG) data.Strengths:Schmidt et al., used a multiverse approach to show that the cardiac influence on this relationship is considerable, by testing a wide range of different analysis parameters (including extensive testing of different frequency ranges assessed to determine the aperiodic fit), algorithms (including different artifact reduction approaches and different aperiodic fitting algorithms), and multiple large datasets to provide conclusions that are robust to the vast majority of potential experimental variations.The study showed that across these different analytical variations, the cardiac contribution to aperiodic activity measured using EEG and MEG is considerable, and likely influences the relationship between aperiodic activity and age to a greater extent than the influence of neural activity.Their findings have significant implications for all future research that aims to assess aperiodic neural activity, suggesting control for the influence of cardiac fields is essential.

We want to thank the reviewer for their thorough engagement with our work and the resultant substantive amount of great ideas both mentioned in the section of Weaknesses and Authors Recommendations below. Their suggestions have sparked many ideas in us on how to move forward in better separating peripheral- from neuro-physiological signals that are likely to greatly influence our future attempts to better extract both cardiac and muscle activity from M/EEG recordings. So we want to thank them for their input, time and effort!

Weaknesses:Figure 4I: The regressions explained here seem to contain a very large number of potential predictors. Based on the way it is currently written, I'm assuming it includes all sensors for both the ECG component and ECG rejected conditions?I'm not sure about the logic of taking a complete signal, decomposing it with ICA to separate out the ECG and non-ECG signals, then including these latent contributions to the full signal back into the same regression model. It seems that there could be some circularity or redundancy in doing so. Can the authors provide a justification for why this is a valid approach?

After observing significant effects both in the MEG_ECG component_ and MEG_ECG rejected_ conditions in similar frequency bands we wanted to understand whether or not these age-related changes are statistically independent. To test this we added both variables as predictors in a regression model (thereby accounting for the influence of the other in relation to age). The regression models we performed were therefore actually not very complex. They were built using only two predictors, namely the data (in a specific frequency range) averaged over channels on which we noticed significant effects in the ECG rejected and ECG components data respectively (Wilkinson notation: *age ~ 1 + ECG rejected + ECG components*). This was also described in the results section stating that: “To see if MEG_ECG rejected_ and MEG_ECG component_ explain unique variance in aging at frequency ranges where we noticed shared effects, we averaged the spectral slope across significant channels and calculated a multiple regression model with MEG_ECG component_ and MEG_ECG rejected_ as predictors for age (to statistically control for the effect of MEG_ECG component_s and MEG_ECG rejected_ on age). This analysis was performed to understand whether the observed shared age-related effects (MEG_ECG rejected_ and MEG_ECG component_) are in(dependent).”

We hope this explanation solves the previous misunderstanding.

I'm not sure whether there is good evidence or rationale to support the statement in the discussion that the presence of the ECG signal in reference electrodes makes it more difficult to isolate independent ECG components. The ICA algorithm will still function to detect common voltage shifts from the ECG as statistically independent from other voltage shifts, even if they're spread across all electrodes due to the referencing montage. I would suggest there are other reasons why the ICA might lead to imperfect separation of the ECG component (assumption of the same number of source components as sensors, non-Gaussian assumption, assumption of independence of source activities).The inclusion of only 32 channels in the EEG data might also have reduced the performance of ICA, increasing the chances of imperfect component separation and the mixing of cardiac artifacts into the neural components, whereas the higher number of sensors in the MEG data would enable better component separation. This could explain the difference between EEG and MEG in the ability to clean the ECG artifact (and perhaps higher-density EEG recordings would not show the same issue).

The reviewer is making a good argument suggesting that our initial assumption that the presence of cardiac activity on the reference electrode influences the performance of the ICA may be wrong. After rereading and rethinking upon the matter we think that the reviewer is correct and that their assumptions for why the ECG signal was not so easily separable from our EEG recordings are more plausible and better grounded in the literature than our initial suggestion. We therefore now highlight their view as a main reason for why the ECG rejection was more challenging in EEG data. However, we also note that understanding the exact reason probably ends up being an empirical question that demands further research stating that:

“Difficulties in removing ECG related components from EEG signals via ICA might be attributable to various reasons such as the number of available sensors or assumptions related to the non-gaussianity of the underlying sources. Further understanding of this matter is highly important given that ICA is the most widely used procedure to separate neural from peripheral physiological sources. ”

In addition to the inability to effectively clean the ECG artifact from EEG data, ICA and other component subtraction methods have also all been shown to distort neural activity in periods that aren't affected by the artifact due to the ubiquitous issue of imperfect component separation (https://doi.org/10.1101/2024.06.06.597688). As such, component subtraction-based (as well as regression-based) removal of the cardiac artifact might also distort the neural contributions to the aperiodic signal, so even methods to adequately address the cardiac artifact might not solve the problem explained in the study. This poses an additional potential confound to the "M/EEG without ECG" conditions.

The reviewer is correct in stating that, if an “artifactual” signal is not always present but appears and disappears (like e.g. eye-blinks) neural activity may be distorted in periods where the “artifactual” signal is absent. However, while this plausibly presents a problem for ocular activity, there is no obvious reason to believe that this applies to cardiac activity. While the ECG signal is non-stationary in nature, it is remarkably more stable than eye-movements in the healthy populations we analyzed (especially at rest). Therefore, the presence of the cardiac “artifact” was consistently present across the entirety of the MEG recordings we visually inspected.

Literature Analysis, Page 23: was there a method applied to address studies that report reducing artifacts in general, but are not specific to a single type of artifact? For example, there are automated methods for cleaning EEG data that use ICLabel (a machine learning algorithm) to delete "artifact" components. Within these studies, the cardiac artifact will not be mentioned specifically, but is included under "artifacts".

The literature analysis was largely performed automatically and solely focussed on ECG related activity as described in the methods section under Literature Analysis, if no ECG related terms were used in the context of artifact rejection a study was flagged as not having removed cardiac activity. This could have been indeed better highlighted by us and we apologize for the oversight on our behalf. We now additionally link to these details stating that:

“However, an analysis of openly accessible M/EEG articles (*NArticles*=279; see Methods - Literature Analysis for further details) that investigate aperiodic activity revealed that only 17.1% of EEG studies explicitly mention that cardiac activity was removed and only 16.5% measure ECG (45.9% of MEG studies removed cardiac activity and 31.1% of MEG studies mention that ECG was measured; see Figure 1EF).”

The reviewer makes a fair point that there is some uncertainty here and our results probably present a lower bound of ECG handling in M/EEG research as, when I manually rechecked the studies that were not initially flagged in studies it was often solely mentioned that “artifacts” were rejected. However, this information seemed too ambiguous to assume that cardiac activity was in fact accounted for. However, again this could have been mentioned more clearly in writing and we apologize for this oversight. Now this is included as part of the methods section *Literature Analysis* stating that:

“All valid word contexts were then manually inspected by scanning the respective word context to ensure that the removal of “artifacts” was related specifically to cardiac and not e.g. ocular activity or the rejection of artifacts in general (without specifying which “artifactual” source was rejected in which case the manuscript was marked as invalid). This means that the results of our literature analysis likely present a lower bound for the rejection of cardiac activity in the M/EEG literature investigating aperiodic activity.”

Statistical inferences, page 23: as far as I can tell, no methods to control for multiple comparisons were implemented. Many of the statistical comparisons were not independent (or even overlapped with similar analyses in the full analysis space to a large extent), so I wouldn't expect strong multiple comparison controls. But addressing this point to some extent would be useful (or clarifying how it has already been addressed if I've missed something).

In the present study we tried to minimize the risk of type 1 errors by several means, such as (A) weakly informative priors, (B) robust regression models and (C) by specifying a region of practical equivalence (ROPE, see Methods Statistical Inference for further Information) to define meaningful effects.

Weakly informative priors can lower the risk of type 1 errors arising from multiple testing by shrinking parameter estimates towards zero (see e.g. Lemoine, 2019). Robust regression models use a Student T distribution to describe the distribution of the data. This distribution features heavier tails, meaning it allocates more probability to extreme values, which in turn minimizes the influence of outliers. The ROPE criterion ensures that only effects exceeding a negligible size are considered meaningful, representing a strict and conservative approach to interpreting our findings (see Kruschke 2018, Cohen, 1988).

Furthermore, and more generally we do not selectively report “significant” effects in the situations in which multiple analyses were conducted on the same family of data (e.g. Figure 2 & 4). Instead we provide joint inference across several plausible analysis options (akin to a specification curve analysis, Simonsohn, Simmons & Nelson 2020) to provide other researchers with an overview of how different analysis choices impact the association between cardiac and neural aperiodic activity.

Lemoine, N. P. (2019). Moving beyond noninformative priors: why and how to choose weakly informative priors in Bayesian analyses. Oikos, 128(7), 912-928.

Simonsohn, U., Simmons, J. P., & Nelson, L. D. (2020). Specification curve analysis. Nature Human Behaviour, 4(11), 1208-1214.

Methods:Applying ICA components from 1Hz high pass filtered data back to the 0.1Hz filtered data leads to worse artifact cleaning performance, as the contribution of the artifact in the 0.1Hz to 1Hz frequency band is not addressed (see Bailey, N. W., Hill, A. T., Biabani, M., Murphy, O. W., Rogasch, N. C., McQueen, B., ... & Fitzgerald, P. B. (2023). RELAX part 2: A fully automated EEG data cleaning algorithm that is applicable to Event-Related-Potentials. Clinical Neurophysiology, result reported in the supplementary materials). This might explain some of the lower frequency slope results (which include a lower frequency limit <1Hz) in the EEG data - the EEG cleaning method is just not addressing the cardiac artifact in that frequency range (although it certainly wouldn't explain all of the results).

We want to thank the reviewer for suggesting this interesting paper, showing that lower high-pass filters may be preferable to the more commonly used >1Hz high-pass filters for detection of ICA components that largely contain peripheral physiological activity. However, the results presented by Bailey et al. contradict the more commonly reported findings by other researchers that >1Hz high-pass filter is actually preferable (e.g. Winkler et al. 2015; Dimingen, 2020 or Klug & Gramann, 2021) and recommendations in widely used packages for M/EEG analysis (e.g. https://mne.tools/1.8/generated/mne.preprocessing.ICA.html). Yet, the fact that there seems to be a discrepancy suggests that further research is needed to better understand which type of high-pass filtering is preferable in which situation. Furthermore, it is notable that all the findings for high-pass filtering in ICA component detection and removal that we are aware of relate to ocular activity. Given that ocular and cardiac activity have very different temporal and spectral patterns it is probably worth further investigating whether the classic 1Hz high-pass filter is really also the best option for the detection and removal of cardiac activity. However, in our opinion this requires a dedicated investigation on its own..

We therefore highlight this now in our manuscript stating that:

“Additionally, it is worth noting that the effectiveness of an ICA crucially depends on the quality of the extracted components(63,64) and even widely suggested settings e.g. high-pass filtering at 1Hz before fitting an ICA may not be universally applicable (see supplementary material of (64)).

Winkler, S. Debener, K. -R. Müller and M. Tangermann, "On the influence of high-pass filtering on ICA-based artifact reduction in EEG-ERP," 2015 37th Annual International Conference of the IEEE Engineering in Medicine and Biology Society (EMBC), Milan, Italy, 2015, pp. 4101-4105, doi: 10.1109/EMBC.2015.7319296.

Dimigen, O. (2020). Optimizing the ICA-based removal of ocular EEG artifacts from free viewing experiments. NeuroImage, 207, 116117.

Klug, M., & Gramann, K. (2021). Identifying key factors for improving ICA‐based decomposition of EEG data in mobile and stationary experiments. European Journal of Neuroscience, 54(12), 8406-8420.

It looks like no methods were implemented to address muscle artifacts. These can affect the slope of EEG activity at higher frequencies. Perhaps the Riemannian Potato addressed these artifacts, but I suspect it wouldn't eliminate all muscle activity. As such, I would be concerned that remaining muscle artifacts affected some of the results, particularly those that included high frequency ranges in the aperiodic estimate. Perhaps if muscle activity were left in the EEG data, it could have disrupted the ability to detect a relationship between age and 1/f slope in a way that didn't disrupt the same relationship in the cardiac data (although I suspect it wouldn't reverse the overall conclusions given the number of converging results including in lower frequency bands). Is there a quick validity analysis the authors can implement to confirm muscle artifacts haven't negatively affected their results?I note that an analysis of head movement in the MEG is provided on page 32, but it would be more robust to show that removing ICA components reflecting muscle doesn't change the results. The results/conclusions of the following study might be useful for objectively detecting probable muscle artifact components: Fitzgibbon, S. P., DeLosAngeles, D., Lewis, T. W., Powers, D. M. W., Grummett, T. S., Whitham, E. M., ... & Pope, K. J. (2016). Automatic determination of EMG-contaminated components and validation of independent component analysis using EEG during pharmacologic paralysis. Clinical neurophysiology, 127(3), 1781-1793.

We thank the reviewer for their suggestion. Muscle activity can indeed be a potential concern, for the estimation of the spectral slope. This is precisely why we used head movements (as also noted by the reviewer) as a proxy for muscle activity. We also agree with the reviewer that this is not a perfect estimate. Additionally, also the riemannian potato would probably only capture epochs that contain transient, but not persistent patterns of muscle activity.

The paper recommended by the reviewer contains a clever approach of using the steepness of the spectral slope (or lack thereof) as an indicator whether or not an independent component (IC) is driven by muscle activity. In order to determine an optimal threshold Fitzgibbon et al. compared paralyzed to temporarily non paralyzed subjects. They determined an expected “EMG-free” threshold for their spectral slope on paralyzed subjects and used this as a benchmark to detect IC’s that were contaminated by muscle activity in non paralyzed subjects.

This is a great idea, but unfortunately would go way beyond what we are able to sensibly estimate with our data for the following reasons. The authors estimated their optimal threshold on paralyzed subjects for EEG data and show that this is a feasible threshold to be applied across different recordings. So for EEG data it might be feasible, at least as a first shot, to use their threshold on our data. However, we are measuring MEG and as alluded to in our discussion section under “Differences in aperiodic activity between magnetic and electric field recordings” the spectral slope differs greatly between MEG and EEG recordings for non-trivial reasons. Furthermore, the spectral slope even seems to also differ across different MEG devices. We noticed this when we initially tried to pool the data recorded in Salzburg with the Cambridge dataset. This means we would need to do a complete validation of this procedure for the MEG data recorded in Cambridge and in Salzburg, which is not feasible considering that we (A) don’t have direct access to one of the recording sites and (B) would even if we had access face substantial hurdles to get ethical approval for the experiment performed by Fitzgibbon et al..

However, we think the approach brought forward by Fitzgibbon and colleagues is a clever way to remove muscle activity from EEG recordings, whenever EMG was not directly recorded. We therefore suggested in the Discussion section that ideally also EMG should be recorded stating that:

“It is worth noting that, apart from cardiac activity, muscle activity can also be captured in (non-)invasive recordings and may drastically influence measures of the spectral slope(72). To ensure that persistent muscle activity does not bias our results we used changes in head movement velocity as a control analysis (see Supplementary Figure S9). However, it should be noted that this is only a proxy for the presence of persistent muscle activity. Ideally, studies investigating aperiodic activity should also be complemented by measurements of EMG. Whenever such measurements are not available creative approaches that use the steepness of the spectral slope (or the lack thereof) as an indicator to detect whether or not e.g. an independent component is driven by muscle activity are promising(72,73). However, these approaches may require further validation to determine how well myographic aperiodic thresholds are transferable across the wide variety of different M/EEG devices.”

**Recommendations for the authors:**

**Reviewer #1 (Recommendations for the authors):**
(1) As outlined above, I recommend rephrasing the last section of the introduction to briefly summarize/introduce all main analysis steps undertaken in the study and why these were done (for example, it is only mentioned that the Cam-CAN dataset was used to study the impact of cardiac on MEG activity although the author used a variety of different datasets). Similarly, I am missing an overview of all main findings in the context of the study goals in the discussion. I believe clarifying the structure of the paper would not only provide a red thread to the reader but also highlight the efforts/strength of the study as described above.

This is a good call! As suggested by the reviewer we now try to give a clearer overview of what was investigated why. We do that both at the end of the introduction stating that: “Using the publicly available Cam-CAN dataset(28,29), we find that the aperiodic signal measured using M/EEG originates from multiple physiological sources. In particular, significant portions of age-related changes in aperiodic activity –normally attributed to neural processes– can be better explained by cardiac activity. This observation holds across a wide range of processing options and control analyses (see Supplementary S1), and was replicable on a separate MEG dataset. However, the extent to which cardiac activity accounts for age-related changes in aperiodic activity varies with the investigated frequency range and recording site. Importantly, in some frequency ranges and sensor locations, age-related changes in neural aperiodic activity still prevail. But does the influence of cardiac activity on the aperiodic spectrum extend beyond age? In a preliminary analysis, we demonstrate that working memory load modulates the aperiodic spectrum of “pure” ECG recordings. The direction of this working memory effect mirrors previous findings on EEG data(5) suggesting that the impact of cardiac activity goes well beyond aging. In sum, our results highlight the complexity of aperiodic activity while cautioning against interpreting it as solely “neural“ without considering physiological influences.”

and at the beginning of the discussion section:

“Difficulties in removing ECG related components from EEG signals via ICA might be attributable to various reasons such as the number of available sensors or assumptions related to the non-gaussianity of the underlying sources. Further understanding of this matter is highly important given that ICA is the most widely used procedure to separate neural from peripheral physiological sources (see Figure 1EF). Additionally, it is worth noting that the effectiveness of an ICA crucially depends on the quality of the extracted components(63,64) and even widely suggested settings e.g. high-pass filtering at 1Hz before fitting an ICA may not be universally applicable (see supplementary material of (64)). “

(2) I found it interesting that the spectral slopes of ECG activity at higher frequency ranges (> 10 Hz) seem mostly related to HRV measures such as fractal and time domain indices and less so with frequency-domain indices. Do the authors have an explanation for why this is the case? Also, the analysis of the HRV measures and their association with aperiodic ECG activity is not explained in any of the method sections.

We apologize for the oversight in not mentioning the HRV analysis in more detail in our methods section. We added a subsection to the Methods section entitled ECG Processing - Heart rate variability analysis to further describe the HRV analyses.

“ECG Processing - Heart rate variability analysis

Heart rate variability (HRV) was computed using the NeuroKit2 toolbox, a high level tool for the analysis of physiological signals. First, the raw electrocardiogram (ECG) data were preprocessed, by highpass filtering the signal at 0.5Hz using an infinite impulse response (IIR) butterworth filter(order=5) and by smoothing the signal with a moving average kernel with the width of one period of 50Hz to remove the powerline noise (default settings of neurokit.ecg.ecg_clean). Afterwards, QRS complexes were detected based on the steepness of the absolute gradient of the ECG signal. Subsequently, R-Peaks were detected as local maxima in the QRS complexes (default settings of neurokit.ecg.ecg_peaks; see (98) for a validation of the algorithm). From the cleaned R-R intervals, 90 HRV indices were derived, encompassing time-domain, frequency-domain, and non-linear measures. Time-domain indices included standard metrics such as the mean and standard deviation of the normalized R-R intervals , the root mean square of successive differences, and other statistical descriptors of interbeat interval variability. Frequency-domain analyses were performed using power spectral density estimation, yielding for instance low frequency (0.04-0.15Hz) and high frequency (0.15-0.4Hz) power components. Additionally, non-linear dynamics were characterized through measures such as sample entropy, detrended fluctuation analysis and various Poincaré plot descriptors. All these measures were then related to the slopes of the low frequency (0.25 – 20 Hz) and high frequency (10 – 145 Hz) aperiodic spectrum of the raw ECG.”

With regards to association of the ECG’s spectral slopes at high frequencies and frequency domain indices of heart rate variability. Common frequency domain indices of heart rate variability fall in the range of 0.01-.4Hz. Which probably explains why we didn’t notice any association at higher frequency ranges (>10Hz).

This is also stated in the related part of the results section:

“In the higher frequency ranges (10 - 145 Hz) spectral slopes were most consistently related to fractal and time domain indices of heart rate variability, but not so much to frequency-domain indices assessing spectral power in frequency ranges < 0.4 Hz.”

(3) Related to the previous point - what is being reflected in the ECG at higher frequency ranges, with regard to biological mechanisms? Results are being mentioned, but not further discussed. However, this point seems crucial because the age effects across the four datasets differ between low and high-frequency slope limits (Figure 2C).

This is a great question that definitely also requires further attention and investigation in general (see also Tereshchenko & Josephson, 2015). We investigated the change of the slope across frequency ranges that are typically captured in common ECG setups for adults (0.05 - 150Hz, Tereshchenko & Josephson, 2015; Kusayama, Wong, Liu et al. 2020). While most of the physiological significant spectral information of an ECG recording rests between 1-50Hz (Clifford & Azuaje, 2006), meaningful information can be extracted at much higher frequencies. For instance, ventricular late potentials have a broader frequency band (~40-250Hz) that falls straight in our spectral analysis window. However, that’s not all, as further meaningful information can be extracted at even higher frequencies (>100Hz). Yet, the exact physiological mechanisms underlying so-called high-frequency QRS remain unclear (HF-QRS; see Tereshchenko & Josephson, 2015; Qiu et al. 2024 for a review discussing possible mechanisms). Yet, at the same time the HF-QRS seems to be highly informative for the early detection of myocardial ischemia and other cardiac abnormalities that may not yet be evident in the standard frequency range (Schlegel et al. 2004; Qiu et al. 2024). All optimism aside, it is also worth noting that ECG recordings at higher frequencies can capture skeletal muscle activity with an overlapping frequency range up to 400Hz (Kusayama, Wong, Liu et al. 2020). We highlight all of this now when introducing this analysis in the results sections as outstanding research question stating that:

“However, substantially less is known about aperiodic activity above 0.4Hz in the ECG. Yet, common ECG setups for adults capture activity at a broad bandwidth of 0.05 - 150Hz(33,34).

Importantly, a lot of the physiological meaningful spectral information rests between 1-50Hz(35), similarly to M/EEG recordings. Furthermore, meaningful information can be extracted at much higher frequencies. For instance, ventricular late potentials have a broader frequency band (~40-250Hz(35)). However, that’s not all, as further meaningful information can be extracted at even higher frequencies (>100Hz). For instance, the so-called high-frequency QRS seems to be highly informative for the early detection of myocardial ischemia and other cardiac abnormalities that may not yet be evident in the standard frequency range(36,37). Yet, the exact physiological mechanisms underlying the high-frequency QRS remain unclear (see (37) for a review discussing possible mechanisms). ”

Tereshchenko, L. G., & Josephson, M. E. (2015). Frequency content and characteristics of ventricular conduction. Journal of electrocardiology, 48(6), 933-937.

Kusayama, T., Wong, J., Liu, X. et al. Simultaneous noninvasive recording of electrocardiogram and skin sympathetic nerve activity (neuECG). Nat Protoc 15, 1853–1877 (2020). https://doi.org/10.1038/s41596-020-0316-6

Clifford, G. D., & Azuaje, F. (2006). Advanced methods and tools for ECG data analysis (Vol. 10). P. McSharry (Ed.). Boston: Artech house.

Qiu, S., Liu, T., Zhan, Z., Li, X., Liu, X., Xin, X., ... & Xiu, J. (2024). Revisiting the diagnostic and prognostic significance of high-frequency QRS analysis in cardiovascular diseases: a comprehensive review. Postgraduate Medical Journal, qgae064.

Schlegel, T. T., Kulecz, W. B., DePalma, J. L., Feiveson, A. H., Wilson, J. S., Rahman, M. A., & Bungo, M. W. (2004, March). Real-time 12-lead high-frequency QRS electrocardiography for enhanced detection of myocardial ischemia and coronary artery disease. In Mayo Clinic Proceedings (Vol. 79, No. 3, pp. 339-350). Elsevier.

(4) Page 10: At first glance, it is not quite clear what is meant by "processing option" in the text. Please clarify.

Thank you for catching this! Upon re-reading this is indeed a bit oblivious. We now swapped “processing options” with “slope fits” to make it clearer that we are talking about the percentage of effects based on the different slope fits.

(5) The authors mention previous findings on age effects on neural 1/f activity (References Nr 5,8,27,39) that seem contrary to their own findings such as e.g., the mostly steepening of the slopes with age. Also, the authors discuss thoroughly why spectral slopes derived from MEG signals may differ from EEG signals. I encourage the authors to have a closer look at these studies and elaborate a bit more on why these studies differ in their conclusions on the age effects. For example, Tröndle et al. (2022, Ref. 39) investigated neural activity in children and young adults, hence, focused on brain maturation, whereas the CamCAN set only considers the adult lifespan. In a similar vein, others report age effects on 1/f activity in much smaller samples as reported here (e.g., Voytek et al., 2015).I believe taking these points into account by briefly discussing them, would strengthen the authors' claims and provide a more fine-grained perspective on aging effects on 1/f.

The reviewer is making a very important point. As age-related differences in (neuro-)physiological activity are not necessarily strictly comparable and entirely linear across different age-cohorts (e.g. age-related changes in alpha center frequency). We therefore, added the suggested discussion points to the discussion section.

“Differences in electric and magnetic field recordings aside, aperiodic activity may not change strictly linearly as we are ageing and studies looking at younger age groups (e.g. <22; 44) may capture different aspects of aging (e.g. brain maturation), than those looking at older subjects (>18 years; our sample). A recent report even shows some first evidence of an interesting putatively non-linear relationship with age in the sensorimotor cortex for resting recordings(59)”

(6) The analysis of the working memory paradigm as described in the outlook-section of the discussion comes as a bit of a surprise as it has not been introduced before. If the authors want to convey with this study that, in general, aperiodic neural activity could be influenced by aperiodic cardiac activity, I recommend introducing this analysis and the results earlier in the manuscript than only in the discussion to strengthen their message.

The reviewer is correct. This analysis really comes a bit out of the blue. However, this was also exactly the intention for placing this analysis in the discussion. As the reviewer correctly noted, the aim was to suggest “that, in general, aperiodic neural activity could be influenced by aperiodic cardiac activity”. We placed this outlook directly after the discussion of “(neuro-)physiological origins of aperiodic activity”, where we highlight the potential challenges of interpreting drug induced changes to M/EEG recordings. So the aim was to get the reader to think about whether age is the only feature affected by cardiac activity and then directly present some evidence that this might go beyond age.

However, we have been rethinking this approach based on the reviewers comments and moved that paragraph to the end of the results section accordingly and introduce it already at the end of the introduction stating that:

“But does the influence of cardiac activity on the aperiodic spectrum extend beyond age? In a preliminary analysis, we demonstrate that working memory load modulates the aperiodic spectrum of “pure” ECG recordings. The direction of this working memory effect mirrors previous findings on EEG data(5) suggesting that the impact of cardiac activity goes well beyond aging.”

(7) The font in Figure 2 is a bit hard to read (especially in D). I recommend increasing the font sizes where necessary for better readability.

We agree with the Reviewer and increased the font sizes accordingly.

(8) Text in the discussion: Figure 3B on page 10 => shouldn't it be Figure 4?

Thank you for catching this oversight. We have now corrected this mistake.

(9) In the third section on page 10, the Figure labels seem to be confused. For example, Figure 4 E is supposed to show "steepening effects", which should be Figure 4B I believe.Please check the figure labels in this section to avoid confusion.

Thank you for catching this oversight. We have now corrected this mistake.

(10) Figure Legend 4 I, please check the figure labels in the text

Thank you for catching this oversight. We have now corrected this mistake.

**Reviewer #3 (Recommendations for the authors):**
I have a number of suggestions for improving the manuscript, which I have divided by section in the following:ABSTRACT:I would suggest re-writing the first sentences to make it easier to read for non-expert readers: "The power of electrophysiologically measured cortical activity decays with an approximately 1/fX function. The slope of this decay (i.e. the spectral exponent, X) is modulated..."

Thank you for the suggestion. We adjusted the sentence as suggested to make it easier for less technical readers to understand that “X” refers to the exponent.

Including the age range that was studied in the abstract could be informative.

Done as suggested.

As an optional recommendation, I think it would increase the impact of the article if the authors note in the abstract that the current most commonly applied cardiac artifact reduction approaches don't resolve the issue for EEG data, likely due to an imperfect ability to separate the cardiac artifact from the neural activity with independent component analysis. This would highlight to the reader that they can't just expect to address these concerns by cleaning their data with typical cleaning methods.I think it would also be useful to convey in the abstract just how comprehensive the included analyses were (in terms of artifact reduction methods tested, different aperiodic algorithms and frequency ranges, and both MEG and EEG). Doing so would let the reader know just how robust the conclusions are likely to be.

This is a brilliant idea! As suggested we added a sentence highlighting that simply performing an ICA may not be sufficient to separate cardiac contributions to M/EEG recordings and refer to the comprehensiveness of the performed analyses.

INTRODUCTION:I would suggest re-writing the following sentence for readability: "In the past, aperiodic neural activity, other than periodic neural activity (local peaks that rise above the "power-law" distribution), was often treated as noise and simply removed from the signal"To something like: "In the past, aperiodic neural activity was often treated as noise and simply removed from the signal e.g. via pre-whitening, so that analyses could focus on periodic neural activity (local peaks that rise above the "power-law" distribution, which are typically thought to reflect neural oscillations).

We are happy to follow that suggestion.

Page 3: please provide the number of articles that were included in the examination of the percentage that remove cardiac activity, and note whether the included articles could be considered a comprehensive or nearly comprehensive list, or just a representative sample.

We stated the exact number of articles in the methods section under *Literature Analysis*. However, we added it to the Introduction on page 3 as suggested by the reviewer. The selection of articles was done automatically, dependent on a list of pre-specified terms and exclusively focussed on articles that had terms related to aperiodic activity in their title (see Literature Analysis). Therefore, I would personally be hesitant in calling it a comprehensive or nearly comprehensive list of the general M/EEG literature as the analysis of aperiodic activity is still relatively niche compared to the more commonly investigated evoked potentials or oscillations. I think whether or not a reader perceives our analysis as comprehensive should be up to them to decide and does not reflect something I want to impose on them. This is exacerbated by the fact that the analysis of neural aperiodic activity has rapidly gained traction over the last years (see Figure 1D orange) and the literature analysis was performed almost 2 years ago and therefore, in my eyes, only represents a glimpse in the rapidly evolving field related to the analysis of aperiodic activity.

Figure 1E-F: It's not completely clear that the "Cleaning Methods" part of the figure indicates just methods to clean the cardiac artifact (rather than any artifact). It also seems that ~40% of EEG studies do not apply any cleaning methods even from within the studies that do clean the cardiac artifact (if I've read the details correctly). This seems unlikely. Perhaps there should be a bar for "other methods", or "unspecified"? Having said that, I'm quite familiar with the EEG artifact reduction literature, and I would be very surprised if ~40% of studies cleaned the cardiac artifact using a different method to the methods listed in the bar graph, so I'm wondering if I've misunderstood the figure, or whether the data capture is incomplete / inaccurate (even though the conclusion that ICA is the most common method is almost certainly accurate).

The cleaning is indeed only focussed on cardiac activity specifically. This was however also mentioned in the caption of Figure 1: “We were further interested in determining which artifact rejection approaches were most commonly used to remove cardiac activity, such as independent component analysis (ICA(22)), singular value decomposition (SVD(23)), signal space separation (SSS(24)), signal space projections (SSP(25)) and denoising source separation (DSS(26)).” and in the methods section under Literature Analysis. However, we adjusted figure 1EF to make it more obvious that the described cleaning methods were only related to the ECG. Aside from using blind source separation techniques such as ICA a good amount of studies mentioned that they cleaned their data based on visual inspection (which was not further considered). Furthermore, it has to be noted that only studies were marked as having separated cardiac from neural activity, when this was mentioned explicitly.

RESULTS:Page 6: I would delete the "from a neurophysiological perspective" clause, which makes the sentence more difficult to read and isn't so accurate (frequencies 13-25Hz would probably more commonly be considered mid-range rather than low or high). Additionally, both frequency ranges include 15Hz, but the next sentence states that the ranges were selected to avoid the knee at 15Hz, which seems to be a contradiction. Could the authors explain in more detail how the split addresses the 15Hz knee?

We removed the “from a neurophysiological perspective” clause as suggested. With regards to the “knee” at ~15Hz I would like to defer the reviewer to Supplementary Figure S1. The Knee Frequency varies substantially across subjects so splitting the data at only 1 exact Frequency did not seem appropriate. Additionally, we found only spurious significant age-related variations in Knee Frequency (i.e. only one out of the 4 datasets; not shown).

Furthermore, we wanted to better connect our findings to our MEG results in Figure 4 and also give the readers a holistic overview of how different frequency ranges in the aperiodic ECG would be affected by age. So to fulfill all of these objectives we decided to fit slopes with respective upper/lower bounds around a range of 5Hz above and below the average 15Hz Knee Frequency across datasets.

The later parts of this same paragraph refer to a vast amount of different frequency ranges, but only the "low" and "high" frequency ranges were previously mentioned. Perhaps the explanation could be expanded to note that multiple lower and upper bounds were tested within each of these low and high frequency windows?

This is a good catch we adjusted the sentence as suggested. We now write: “.. slopes were fitted individually to each subject's power spectrum in several lower (0.25 – 20 Hz) and higher (10-145 Hz) frequency ranges.”

The following two sentences seem to contradict each other: "Overall, spectral slopes in lower frequency ranges were more consistently related to heart rate variability indices(> 39.4% percent of all investigated indices)" and: "In the lower frequency range (0.25 - 20Hz), spectral slopes were consistently related to most measures of heart rate variability; i.e. significant effects were detected in all 4 datasets (see Figure 2D)." (39.4% is not "most").

The reviewer is correct in stating that 39.4% is not most. However, the 39.4% is the lowest bound and only refers to 1 dataset. In the other 3 datasets the percentage of effects was above 64% which can be categorized as “most” i.e. above 50%. We agree that this was a bit ambiguous in the sentence so we added the other percentages as well as a reference to Figure 2D to make this point clearer.

Figure 2D: it isn't clear what the percentages in the semi-circles reflect, nor why some semi-circles are more full circles while others are only quarter circles.

The percentages in the semi-circles reflect the amount of effects (marked in red) and null effects (marked in green) per dataset, when viewed as average across the different measures of HRV. Sometimes less effects were found for some frequency ranges resulting in quarters instead of semi circles.

Page 8: I think the authors could make it more clear that one of the conditions they were testing was the ECG component of the EEG data (extracted by ICA then projected back into the scalp space for the temporal response function analysis).

As suggested by the reviewer we adjusted our wording and replaced the arguably a bit ambiguous “... projected back separately” with “... projected back into the sensor space”. We thank the reviewer for this recommendation, as it does indeed make it easier to understand the procedure.

“After pre-processing (see Methods) the data was split in three conditions using an ICA(22). Independent components that were correlated (at *r* > 0.4; see Methods: MEG/EEG Processing - pre-processing) with the ECG electrode were either not removed from the data (Figure 3ABCD - blue), removed from the data (Figure 2ABCD - orange) or projected back into the sensor space (Figure 3ABCD - green).”

Figure 4A: standardized beta coefficients for the relationship between age and spectral slope could be noted to provide improved clarity (if I'm correct in assuming that is what they reflect).

This was indeed shown in Figure 4A and noted in the color bar as “average beta (standardized)”. We do not specifically highlight this in the text, because the exact coefficients would depend on both on the analyzed frequency range and the selected electrodes.

Figure 4I: The regressions explained at this point seems to contain a very large number of potential predictors, as I'm assuming it includes all sensors for both the ECG component and ECG rejected conditions? (if that is not the case, it could be explained in greater detail). I'm also not sure about the logic of taking a complete signal, decomposing it with ICA to separate out the ECG and non-ECG signals, then including them back into the same regression model. It seems that there could be some circularity or redundancy in doing so. However, I'm not confident that this is an issue, so would appreciate the authors explaining why it this is a valid approach (if that is the case).

After observing significant effects both in the MEG_ECG component_ and MEG_ECG rejected_ conditions in similar frequency bands we wanted to understand whether or not these age-related changes are statistically independent. To test this we added both variables as predictors in a regression model (thereby accounting for the influence of the other in relation to age). The regression models we performed were therefore actually not very complex. They were built using only two predictors, namely the data (in a specific frequency range) averaged over channels on which we noticed significant effects in the ECG rejected and ECG components data respectively (Wilkinson notation: *age ~ 1 + ECG rejected + ECG components*). This was also described in the results section stating that: “To see if MEG_ECG rejected_ and MEG_ECG component_ explain unique variance in aging at frequency ranges where we noticed shared effects, we averaged the spectral slope across significant channels and calculated a multiple regression model with MEG_ECG component_ and MEG_ECG rejected_ as predictors for age (to statistically control for the effect of MEG_ECG component_s and MEG_ECG rejected_ on age). This analysis was performed to understand whether the observed shared age-related effects (MEG_ECG rejected_ and MEG_ECG component_) are in(dependent).”

We hope this explanation solves the previous misunderstanding.

The explanation of results for relationships between spectral slopes and aging reported in Figure 4 refers to clusters of effects, but the statistical inference methods section doesn't explain how these clusters were determined.

The wording of “cluster” was used to describe a “category” of effects e.g. null effects. We changed the wording from “cluster” to “category” to make this clearer stating now that: “This analysis, which is depicted in Figure 4, shows that over a broad amount of individual fitting ranges and sensors, aging resulted in a steepening of spectral slopes across conditions (see Figure 4E) with “steepening effects” observed in 25% of the processing options in MEG_ECG not rejected_ , 0.5% in MEG_ECG rejected_, and 60% for MEG_ECG components_. The second largest category of effects were “null effects” in 13% of the options for MEG_ECG not rejected_ , 30% in MEG_ECG rejected_, and 7% for MEG_ECG components_. ”

Page 12: can the authors clarify whether these age related steepenings of the spectral slope in the MEG are when the data include the ECG contribution, or when the data exclude the ECG? (clarifying this seems critical to the message the authors are presenting).

We apologize for not making this clearer. We now write: “This analysis also indicates that a vast majority of observed effects irrespective of condition (ECG components, ECG not rejected, ECG rejected) show a steepening of the spectral slope with age across sensors and frequency ranges.”

Page 13: I think it would be useful to describe how much variance was explained by the MEG-ECG rejected vs MEG-ECG component conditions for a range of these analyses, so the reader also has an understanding of how much aperiodic neural activity might be influenced by age (vs if the effects are really driven mostly by changes in the ECG).

With regards to the explained variance I think that the very important question of how strong age influences changes in aperiodic activity is a topic better suited for a meta analysis. As the effect sizes seems to vary largely depending on the sample e.g. for EEG in the literature results were reported at r=-0.08 (Cesnaite et al. 2023), r=-0.26 (Cellier et al. 2021), r=-0.24/r=-0.28/r=-0.35 (Hill et al. 2022) and r=0.5/r=0.7 (Voytek et al. 2015). I would defer the reader/reviewer to the standardized beta coefficients as a measure of effect size in the current study that is depicted in Figure 4A.

Cellier, D., Riddle, J., Petersen, I., & Hwang, K. (2021). The development of theta and alpha neural oscillations from ages 3 to 24 years. Developmental cognitive neuroscience, 50, 100969.

Cesnaite, E., Steinfath, P., Idaji, M. J., Stephani, T., Kumral, D., Haufe, S., ... & Nikulin, V. V. (2023). Alterations in rhythmic and non‐rhythmic resting‐state EEG activity and their link to cognition in older age. NeuroImage, 268, 119810.

Hill, A. T., Clark, G. M., Bigelow, F. J., Lum, J. A., & Enticott, P. G. (2022). Periodic and aperiodic neural activity displays age-dependent changes across early-to-middle childhood. Developmental Cognitive Neuroscience, 54, 101076.

Voytek, B., Kramer, M. A., Case, J., Lepage, K. Q., Tempesta, Z. R., Knight, R. T., & Gazzaley, A. (2015). Age-related changes in 1/f neural electrophysiological noise. Journal of Neuroscience, 35(38), 13257-13265.

Also, if there are specific M/EEG sensors where the 1/f activity does relate strongly to age, it would be worth noting these, so future research could explore those sensors in more detail.

I think it is difficult to make a clear claim about this for MEG data, as the exact location or type of the sensor may differ across manufacturers. Such a statement could be easier made for source projected data or in case EEG electrodes were available, where the location would be normed eg. according to the 10-20 system.

DISCUSSION:Page 15: Please change the wording of the following sentence, as the way it is currently worded seems to suggest that the authors of the current manuscript have demonstrated this point (which I think is not the case): "The authors demonstrate that EEG typically integrates activity over larger volumes than MEG, resulting in differently shaped spectra across both recording methods."

Apologies for the oversight! The reviewer is correct we in fact did not show this, but the authors of the cited manuscript. We correct the sentence as suggested stating now that:

“Bénar et al. demonstrate that EEG typically integrates activity over larger volumes than MEG, resulting in differently shaped spectra across both recording methods.”

Page 16: The authors mention the results can be sensitive to the application of SSS to clean the MEG data, but not ICA. I think it would be sensitive to the application of either SSS or ICA?

This is correct and actually also supported by Figure S7, as differences in ICA thresholds affect also the detection of age-related effects. We therefore adjusted the related sentences stating now that:

“ In case of the MEG signal this may include the application of Signal-Space-Separation algorithms (SSS(24,55)), different thresholds for ICA component detection (see Figure S7), high and low pass filtering, choices during spectral density estimation (window length/type etc.), different parametrization algorithms (e.g. IRASA vs FOOOF) and selection of frequency ranges for the aperiodic slope estimation.”

It would be worth clarifying that the linked mastoid re-reference alone has been proposed to cancel out the ECG signal, rather than that a linked-mastoid re-reference improves the performance of the ICA separation (which could be inferred by the explanation as it's currently written).

This is correct and we adjusted the sentence accordingly! Stating now that:

“ Previous work(12,56) has shown that a linked mastoid reference alone was particularly effective in reducing the impact of ECG related activity on aperiodic activity measured using EEG. “

The issue of the number of EEG channels could probably just be noted as a potential limitation, as could the issue of neural activity being mixed into the ECG component (although this does pose a potential confound to the M/EEG without ECG condition, I suspect it wouldn't be critical).

This is indeed a very fair point as a higher amount of electrodes would probably make it easier to better isolate ECG components in the EEG, which may be the reason why the separation did not work so well in our case. However, this is ultimately an empirical question so we highlighted it in the discussion section stating that: “Difficulties in removing ECG related components from EEG signals via ICA might be attributable to various reasons such as the number of available sensors or assumptions related to the non-gaussianity of the underlying sources. Further understanding of this matter is highly important given that ICA is the most widely used procedure to separate neural from peripheral physiological sources. ”

OUTLOOK:Page 19: Although there has been a recent trend to control for 1/f activity when examining oscillatory power, recent research suggests that this should only be implemented in specific circumstances, otherwise the correction causes more of a confound than the issue does. It might be worth considering this point with regards to the final recommendation in the Outlook section: Brake, N., Duc, F., Rokos, A., Arseneau, F., Shahiri, S., Khadra, A., & Plourde, G. (2024). A neurophysiological basis for aperiodic EEG and the background spectral trend. Nature Communications, 15(1), 1514.

We want to thank the reviewer for recommending this very interesting paper! The authors of said paper present compelling evidence showing that, while peak detection above an aperiodic trend using methods like FOOOF or IRASA is a prerequisite to determine the presence of oscillatory activity, it’s not necessarily straightforward to determine which detrending approach should be applied to determine the actual power of an oscillation. Furthermore, the authors suggest that wrongfully detrending may cause larger errors than not detrending at all. We therefore added a sentence stating that: “However, whether or not periodic activity (after detection) should be detrended using approaches like FOOOF or IRASA still remains disputed, as incorrectly detrending the data may cause larger errors than not detrending at all(75).”

RECOMMENDATIONS:Page 20: "measure and account for" seems like it's missing a word, can this be re-written so the meaning is more clear?

Done as suggested. The sentence now states: “To better disentangle physiological and neural sources of aperiodic activity, we propose the following steps to (1) measure and (2) account for physiological influences.”

I would re-phrase "doing an ICA" to "reducing cardiac artifacts using ICA" (this wording could be changed in other places also).

I do not like to describe cardiac or ocular activity as artifactual per se. This is also why I used hyphens whenever I mention the word “artifact” in association with the ECG or EOG. However, I do understand that the wording of “doing an ICA” is a bit sloppy. We therefore reworded it accordingly throughout the manuscript to e.g. “separating cardiac from neural sources using an ICA” and “separating physiological from neural sources using an ICA”.

I would additionally note that even if components are identified as unambiguously cardiac, it is still likely that neural activity is mixed in, and so either subtracting or leaving the component will both be an issue (https://doi.org/10.1101/2024.06.06.597688). As such, even perfect identification of whether components are cardiac or not would still mean the issue remains (and this issue is also consistent across a considerable range of component based methods). Furthermore, current methods including wavelet transforms on the ICA component still do not provide good separation of the artifact and neural activity.

This is definitely a fair point and we also highlight this in our recommendations under 3 stating that:

“However, separating physiological from neural sources using an ICA is no guarantee that peripheral physiological activity is fully removed from the cortical signal. Even more sophisticated ICA based methods that e.g. apply wavelet transforms on the ICA components may still not provide a good separation of peripheral physiological and neural activity76,77. This turns the process of deciding whether or not an ICA component is e.g. either reflective of cardiac or neural activity into a challenging problem. For instance, when we only extract cardiac components using relatively high detection thresholds (e.g. r > 0.8), we might end up misclassifying residual cardiac activity as neural. In turn, we can’t always be sure that using lower thresholds won’t result in misinterpreting parts of the neural effects as cardiac. Both ways of analyzing the data can potentially result in misconceptions.”

Castellanos, N. P., & Makarov, V. A. (2006). Recovering EEG brain signals: Artifact suppression with wavelet enhanced independent component analysis. Journal of neuroscience methods, 158(2), 300-312.

Bailey, N. W., Hill, A. T., Godfrey, K., Perera, M. P. N., Rogasch, N. C., Fitzgibbon, B. M., & Fitzgerald, P. B. (2024). EEG is better when cleaning effectively targets artifacts. bioRxiv, 2024-06.

METHODS:Pre-processing, page 24: I assume the symmetric setting of fastica was used (rather than the deflation setting), but this should be specified.

Indeed the reviewer is correct, we used the standard setting of fastICA implemented in MNE python, which is calling the FastICA implementation in sklearn that is per default using the “parallel” or symmetric algorithm to compute an ICA. We added this information to the text accordingly, stating that:

“For extracting physiological “artifacts” from the data, 50 independent components were calculated using the *fastica* algorithm(22) (implemented in MNE-Python *version 1.2*; with the parallel/symmetric setting; note: 50 components were selected for MEG for computational reasons for the analysis of EEG data no threshold was applied).”

Temporal response functions, page 26: can the authors please clarify whether the TRF is computed against the ECG signal for each electrode or sensory independently, or if all electrodes/sensors are included in the analysis concurrently? I'm assuming it was computed for each electrode and sensory separately, since the TRF was computed in both the forward and backwards direction (perhaps the meaning of forwards and backwards could be explained in more detail also - i.e. using the ECG to predict the EEG signal, or using the EEG signal to predict the ECG signal?).

A TRF can also be conceptualized as a multiple regression model over time lags. This means that we used all channels to compute the forward and backward models. In the case of the forward model we predicted the signal of the M/EEG channels in a multivariate regression model using the ECG electrode as predictor. In case of the backward model we predicted the ECG electrode based on the signal of all M/EEG channels. The forward model was used to depict the time window at which the ECG signal was encoded in the M/EEG recording, which appears at 0 time lags indicating volume conduction. The backward model was used to see how much information of the ECG was decodable by taking the information of all channels.

We tried to further clarify this approach in the methods section stating that:

“We calculated the same model in the forward direction (encoding model; i.e. predicting M/EEG data in a multivariate model from the ECG signal) and backward direction (decoding model; i.e. predicting the ECG signal using all M/EEG channels as predictors).”

Page 27: the ECG data was fit using a knee, but it seems the EEG and MEG data was not.Does this different pose any potential confound to the conclusions drawn? (having said this, Figure S4 suggests perhaps a knee was tested in the M/EEG data, which should perhaps be explained in the text also).

This was indeed tested in a previous review round to ensure that our results are not dependent on the presence/absence of a knee in the data. We therefore added figure S4, but forgot to actually add a description in the text. We are sorry for this oversight and added a paragraph to S1 accordingly:

“Using FOOOF(5), we also investigated the impact of different slope fitting options (fixed vs. knee model fits) on the aperiodic age relationship (see Supplementary Figure S4). The results that we obtained from these analyses using FOOOF offer converging evidence with our main analysis using IRASA.”

Page 32: my understanding of the result reported here is that cleaning with ICA provided better sensitivity to the effects of age on 1/f activity than cleaning with SSS. Is this accurate? I think this could also be reported in the main manuscript, as it will be useful to researchers considering how to clean their M/EEG data prior to analyzing 1/f activity.

The reviewer is correct in stating that we overall detected slightly more “significant” effects, when not additionally cleaning the data using SSS. However, I am a bit wary of recommending omitting the use of SSS maxfilter solely based on this information. It can very well be that the higher quantity of effects (when not employing SSS maxfilter) stems from other physiological sources (e.g. muscle activity) that are correlated with age and removed when applying SSS maxfiltering. I think that just conditioning the decision of whether or not maxfilter is applied based on the amount or size of effects may not be the best idea. Instead I think that the applicability of maxfilter for research questions related to aperiodic activity should be the topic of additional methodological research. We therefore now write in Text S1:

“Considering that we detected less and weaker aperiodic effects when using SSS maxfilter is it advisable to omit maxfilter, when analyzing aperiodic signals? We don’t think that we can make such a judgment based on our current results. This is because it's unclear whether or not the reduction of effects stems from an additional removal of peripheral information (e.g. muscle activity; that may be correlated with aging) or is induced by the SSS maxfiltering procedure itself. As the use of maxfilter in detecting changes of aperiodic activity was not subject of analysis that we are aware of, we suggest that this should be the topic of additional methodological research.”

Page 39, Figure S6 and Figure S8: Perhaps the caption could also briefly explain the difference between maxfilter set to false vs true? I might have missed it, but I didn't gain an understanding of what varying maxfilter would mean.

Figure S6 shows the effect of ageing on the spectral slope averaged across all channels. The maxfilter set to false in (AB) means that no maxfiltering using SSS was performed vs. in (CD) where the data was additionally processed using the SSS maxfilter algorithm. We now describe this more clearly by writing in the caption:

“Supplementary Figure S6: Age-related changes in aperiodic brain activity are most prominent on explained by cardiac components irrespective of maxfiltering the data using signal space separation (SSS) or not (AC) Age was used to predict the spectral slope (fitted at 0.1-145Hz) averaged across sensors at rest in three different conditions ECG components not rejected [blue], ECG components rejected [orange], ECG components only [green].”